# Unlock the Intermittent Control Ability of Model Free Reinforcement Learning

**Jiashun Liu, Jianye Hao**∗**& Xiaotian Hao**
College of Intelligence and Computing
Tianjin University
China

**Yi Ma**
School of Computer and Information Technology
Shanxi University
China

**YAN ZHENG**
Tianjin University
China
yanzheng@tju.edu.cn

**Yujing Hu & Tangji Lv**
FUXI AI Laboratory
NetEase
China

## Abstract

Intermittent control problems are common in real world. The interactions between the decision maker and the executor can be discontinuous (intermittent) due to various types of interruptions, e.g. unstable communication channel. Due to intermittent interaction, agents are unable to acquire the state sent by the executor and cannot transmit actions to the executor within a period of time step, i.e. bidirectional blockage, which may lead to inefficiencies of reinforcement learning policies and prevent the executors from completing the task. Such problem is not well studied in the RL community. In this paper, we model Intermittent Control Problem as an Intermittent Control Markov Decision Process, i.e., agents are expected to generate action sequences corresponding to the unavailable states and transmit them before disabling interactions to ensure the smooth and effective motion of executors. However, directly generating multiple future actions in the original action space has unnatural motion issue and exploration difficulty. We propose **M**ulti-step **A**ction **R**epre**S**entation (**MARS**), which encodes a sequence of actions from the original action space to a compact and decodable latent space. Then based on the latent action sequence representation, the mainstream RL methods can be easily optimized to learn a smooth and efficient motion policy. Extensive experiments on simulation tasks and real-world robotic grasping tasks show that MARS significantly improves the learning efficiency and final performances compared with existing baselines.

## 1 Introduction

In recent years, the field of **d**eep **r**einforcement **l**earning (DRL) has witnessed striking empirical achievements in a variety of Markov Decision Process (MDP) problems [Mnih et al., 2013, Kaufmann et al., 2023] and has been successfully applied to many real-time control tasks [Mahmood et al., 2018, Lee et al., 2020]. In real-time control, "interaction" plays a crucial role. At each time step, the decision maker obtains observations from the environment and feeds actions back to the action executor through real-time interactions. Thus, in an ideal MDP setting, the interaction should be continuous. However, in many scenarios, interactions become intermittent due to limitations of realistic conditions [Jiang et al., 2021] or communication interruption (e.g., unstable network) [Dong et al., 2009]. Due to

---

∗Correspondence to: Jianye Hao <jianye.hao@tju.edu.cn>

bidirectional communication blockage caused by intermittent interaction, agents (decision makers) are unable to acquire the state sent by the executor and cannot transmit actions to the executor within a period of time step (as shown in Figure 1 (top). The sparse observations resulting from this phenomenon can make normal step-by-step decisions unstable. Thus, directly deploying existing DRL algorithms could make the action executor abruptly stop (can not make decision when the current state is unavailable) or move abnormally when the environment suddenly changes [Sutton and Barto, 2018].

Typically, there are two types of intermittent interactions. Illustrations are shown in Figure 1(bottom). ❶ **Fixed interaction interval**: in numerous real-world robot scenarios, interacting with the environment is often time-consuming and costly [Liu et al., 2020]. For example, before the robotic arm performs its next actions, it has to first halt its current operations, move to a designated position to capture an image, and then use a specific technique to extract features and provide them to the decision maker. To ensure reliable operation and stable movement, we usually set a fixed interaction interval for the robot arm [Jiang et al., 2021, Bonarini, 2020]. ❷ **Random interaction interval**: unexpected interaction intervals may occur due to unstable communication channels and loss of wireless signals [Li et al., 2016]. For example, in real-time strategy games, the

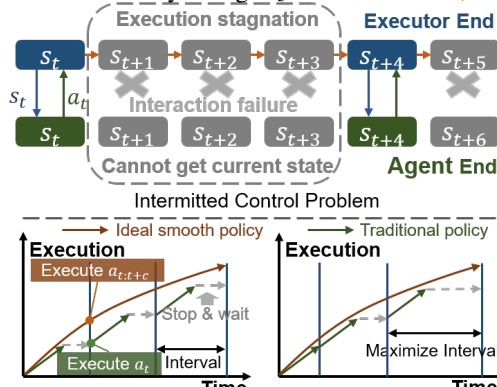

Figure 1: Detail of Intermitted Control (top part). Two types of intermittent interactions (bottom part). For both cases, we can generate a sequence of actions in advance for the next $c$ states based on the current state $s_t$ to make the control more smooth and robust.

decision end remotely controls the terminal non-player characters (NPC) [Zheng et al., 2019], the interaction interval between the decision end and NPC terminals may become random [Wong et al., 2021] due to above reasons, which may cause the NPC to be stuck and disconnected from the changed environment, reducing the player's experience.

The essence of addressing the two aforementioned intermittent control tasks is identical, that is, to achieve intensive control under sparse interaction (or observation) to ensure the effective and smooth movement of the executor, ultimately leading to efficient task completion. To this end, we introduce Intermittent-MDP (refer to Sec.3 for further elaboration) to model the above two settings in a unified manner, that is, the agent is expected to decide on a sequence of actions based on the current state, covering a suitable number of time steps, to maintain smooth and efficient motion of the executor between the two interactions. The most direct approach would be to employ model-based reinforcement learning (MBRL) methods with multi-step decision-making capabilities, such as Dreamer [Hafner et al., 2023] and TD-MPC [Hansen et al., 2022]. Regrettably, the form of multi-step decision makes the error of the dynamic model accumulate and then make the policies suboptimal. In addition, the high demand for high-quality data and the complexity of constructing dynamic models makes MBRL deployment costly in real-world scenarios [Janner et al., 2019] (detailed experimental analysis in Appendix C.1). Instead, we sought to unlock the multi-step decision-making ability of the model-free DRL approach, e.g. TD3 [Fujimoto et al., 2018] and PPO [Schulman et al., 2017].

The most simple method for model-free DRL to alleviate the intermittent interaction issue is using frameskip (also commonly known as 'action-repeat') [Kalyanakrishnan et al., 2021], where the same action (usually the last action) is repeated during a fixed interval which is often used in Atari [Braylan et al., 2015]. However, longer frameskip will lead to mechanized motion, making it impossible for the agent to change actions at key states and thus resulting in suboptimal policies. Another way is to let the RL algorithms make up-front decisions (advance decisions) for future steps according to the current state or the received delayed state. Only actions for the correct time steps will be executed. Compared to frameskip, these methods can improve action diversity. However, directly increasing the horizon of decision making will lead to action space explosion and increase the difficulty of policy optimization [Chen et al., 2021].

In this paper, we propose Multi-step Action RepreSentation (MARS), which is the first plugin method for DRL algorithms to solve intermittent control tasks, significantly reducing the difficulty of multi-step policy training while ensuring the flexibility and diversity. The high level idea is shown in Figure 2. MARS constructs a compact and decodable low dimensional latent action

space for the original multi-step actions, the latent action dimension does not explode as the steps number increases. Based on this latent action space agent can learn a stable latent policy. The latent action selected by the policy can be reconstructed into original multi-step actions by the decoder. Specifically, MARS relies on a conditional Variational Auto-encoder (c-VAE) [Sohn et al., 2015] to construct the latent representation space for the associated multi-step actions. Intuitively, an efficient latent action space should have two characteristics: ❶ **Decision space simplification**: the combination number of long-sequence actions is huge, especially in the continuous action space, which can complicate the decision space and potentially result in suboptimal policies. We restrict the decision space to a relatively small subspace by introducing the concept of action transfer scale $\upsilon$ and taking it as the condition term of our VAE, i.e. using $\upsilon$ to implicit segment the latent space [Sohn et al., 2015]. ❷ **Dynamic semantic smoothness**, a characteristic of the original action space, which means latent actions that have similar impacts on the environment should be close together in latent spaces. Dynamic semantic smooth action space can retain the utility of adding Gaussian noise perturbations to the policy during training: While ensuring that the impacts of initial latent action chosen by the policy and latent action after disturbance on the environment is similar in high probability, some uncertainty is also retained that the disturbed action may explore rare actions that have unknown impacts on the environment, which helps maintain a balance between exploration and exploitation [Schwarzer et al., 2020]. We utilize unsupervised environmental dynamics to learn a predictive representation of dynamics for improving semantic smoothness. In practice, we adding an additional dynamic residual prediction module for VAE.

Our contributions are summarized as follows: ➤ We propose the first plugin method and DRL framework for solving intermittent control tasks via multi-step action representation learning. ➤ We provide two modules to improve the effectiveness and smoothness of the learned multi-step latent space. ➤ Our method outperforms baselines on both real-world fixed interaction interval robotic control tasks and random interaction interval simulation control tasks.

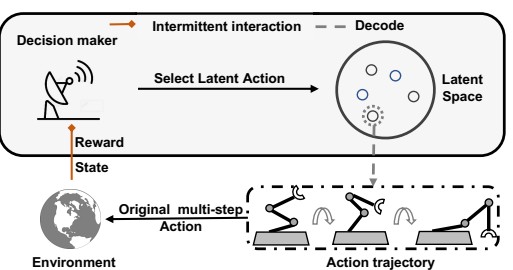

Figure 2: Conceptual overview of MARS.

## 2 Background

**Markov Decision Process (MDP).** A MDP can be represented as a tuple: $(\mathcal{S}, \mathcal{A}, \mathcal{P}, \mathcal{R}, \gamma, T)$, where $\mathcal{S}$ denotes the state set, $\mathcal{A}$ denotes an action set, $\mathcal{P}$ is the transition function: $\mathcal{S} \times \mathcal{A} \times \mathcal{S} \rightarrow [0, 1]$ and $\mathcal{R}$ is the reward function: $\mathcal{S} \times \mathcal{A} \rightarrow \mathbb{R}$. $\gamma \in [0, 1)$ is a discount factor and $T$ is the decision horizon. The goal is to optimize the agent's policy to maximize the expected discounted cumulative reward.

**Variational Auto-Encoder.** The variational auto-encoder (VAE) is a directed graphical model with certain types of latent variables, such as Gaussian latent variables. A generative process of the VAE is as follows: a set of latent variable $z$ is generated from the prior distribution $p_\theta(z)$ and the data x is generated by the generative distribution $p_\theta(x|z)$ conditioned on $z: z \sim p_\theta(z), x \sim p_\theta(x|z)$. In general, parameter estimation of directed graphical models is often challenging due to intractable posterior inference. However, the parameters of the VAE can be estimated efficiently using the stochastic gradient variational Bayes (SGVB) framework, where the variational lower bound of the log-likelihood is used as a surrogate objective function. In this framework, a proposal distribution $q_\theta(x|z)$, is introduced to approximate the true posterior $p_\theta(x|z)$. MLPs are used to model the recognition and the generation models. Based on the Gaussian latent assumption, the first term of Eq.1 can be marginalized. The second term can be approximated by drawing samples $z^{(l)}(l = 1, ..., L)$ by the proposal distribution $q_\theta(x|z)$. The empirical objective of the VAE is written as:

$$L_{VAE}(\phi, \psi) = \frac{1}{L} \sum_\theta (x|z^{(l)}) - KL\big(q_\phi(z|x)||N(0, I)\big) \tag{1}$$

# 3   Intermittent Control Markov Decision Process (Intermittent-MDP)

In this section, we introduce the Intermittent Control Markov Decision Process (Intermittent-MDP) to model both fixed and random interaction interval control problems. The objective of Intermittent-MDP is to train the policy to decide on an effective action sequence $u$ based on the current state $s$, which should be longer than or equal to the maximum interaction interval $c$, thus ensuring smooth and efficient motion of executor between the two interactions.

> **Definition of Intermittent-MDP.** Intermittent-MDP ($\mathcal{M}$) can be represented as a tuple: $\mathcal{M} \triangleq \langle \mathcal{S}, \mathcal{U}, \mathcal{R}, \mathcal{K}, \mu_0 \rangle$. $S \in \mathbb{R}^n$ is the state space, $\mathcal{U}$ is the set of action sequence $u$ that the policy can select. and $\mu_0$ denotes the initial state distribution. The reward is defined as $\mathcal{R}$, $\mathcal{K}$ is the multi-step transition function.

Different from normal MDP in which the agent makes a decision $a_t$ according to the current state $s_t$, intermittent-MDP requires the agent to generate an action sequence $u_t = \{a_t, ..., a_{t+c}\}$ according to the received state $s_t$ ($a_t$ is the single step action at timestep $t$). The cumulative rewards of executed actions can be received in the next interaction. Policy $\pi$ takes current state to select action sequence $u_t = \{a_t, ..., a_{t+c}\}$. When the interaction interval is present for $j$ timesteps, the executor uses $j$ single-step actions in the action sequence $u$ to maintain the motion ($j \in \{0, 1, ..., c\}$). When the interaction is continuous, the executor uses the first single-step action in $u$ each time (degenerated to the transition function $\mathcal{P}$ in normal MDP). Thus, the environment transition function $\mathcal{K}$ is defined as:

$$\mathcal{K}(s_{t+j}|s_t, u_t) = \Pi_{i=t}^{t+j-1} \mathcal{P}(s_{i+1}|s_i, a_i) \pi(u_t|s_t) \ \ j \in \{0, 1, ..., c\}, \ a_i \in u_t. \tag{2}$$

If the interval is random and we only know the maximum number of interval step $c$, Intermittent-MDP is represented as *random Intermittent-MDP*. When the interval time is fixed, Intermittent-MDP represents a fixed interaction interval problem, i.e., *fixed Intermittent-MDP*.

# 4   Multi-step Action Representation

In this section, we introduce Multi-step Action Representation (MARS), a novel framework that can map long action sequences into an efficient latent space. We hypothesize that multi-step actions reside on a homogeneous manifold. To learn the multi-step action representation, we employ VAE to constructing a compact and decodable latent representation space to approximate such a manifold.

To align our latent action space with the two characteristics mentioned in Sec.1, i.e. ① Decision space simplification ② Dynamic semantic smoothness. In Sec.4.1, we introduce the concept of action transition scale and encode this as a conditional aid in VAE training to simplify the decision space of the policy. In Sec.4.2, we demonstrate how state residual prediction can be leveraged to smooth the semantics of dynamic transitions in latent action spaces, and in Sec.4.3, we illustrate the combination of MARS with DRL, as well as the extension of our approach to random intermittent-MDP.

## 4.1   Scale-Conditioned Multi-step Action Encoding and Decoding

Although VAE has been proven to be able to construct a compact and effective latent action space for normal MDP with state $s$ as the prior condition [Li et al., 2021], due to the complex combinatorial nature of multi-step actions in Intermittent-MDP, the policy still struggles to explore the optimal actions in the vast action space. Therefore, to address this challenge, we introduce the concept of **action transition scale** $\upsilon$ as a conditional term in VAE, to constructs the multi-step latent action space $\mathcal{Z}$. $\upsilon$ stands for a description of the motion style, it represents the accumulation of action change scales within each action sequence $u$. By determining $\upsilon$, we can constrain the latent action $z \in Z$ chosen by the policy within a related subspace [Sohn et al., 2015] in which all candidate action sequences have the same motion style in terms of $\upsilon$, thus reducing the difficulty of exploration. We choose $\upsilon$ as a conditional term due to its task-specific nature. For instance, in robot scenarios, to ensure the balance of the robot and prevent joint damage, the scale of action changes is typically small. In some obstacle avoidance tasks, however, the magnitude of action changes can be significant when the agent encounters suddenly approaching obstacles. Once the policy roughly determines the appropriate action transfer scale for the current state, it can make more suitable decisions, making the policy more efficient and controllable in the complex action space. Notably $\upsilon$ does not need to be

set manually but is selected adaptively by the policy through training, which is described in detail in Sec.4.3. We formulate $\upsilon$ as follows:

$$\upsilon_{u_t} = \frac{\sum_{i=t}^{c-1} \|a_{i+1} - a_i\|}{(c-1) \times B}. \tag{3}$$

$c$ is the maximum interval and $B$ denotes the upper limit of action change. The numerator part represents the total absolute difference between consecutive actions, used to assess the transition magnitude of $u_t$. Eq.3 normalizes $\upsilon_{u_t}$ to $[0, 1]$.

Given an action sequence $u_t$ and the corresponding states $s_{t:t+c}$, our encoder $q_\phi(z_{u_t}|u_t, s_{t:t+c}, \upsilon_{u_t})$ parameterized by $\phi$ takes $s_{t:t+c}$ and the action transition scale $\upsilon_{u_t}$ as conditions to build a multi-step latent action space, and maps the action sequence $u_t$ into the latent variable $z_{u_t} \in \mathbb{R}^{d_1}$ ($d_1$ denotes the dimension of $z_{u_t}$). The decoder $p_\psi(\hat{u}_t|z_{u_t}, s_t, \upsilon_{u_t})$ parameterized by $\psi$ then reconstructs the multi-step actions $u_t$ from $z_{u_t}$. We employ a Gaussian latent distribution $N(\mu_z, \sigma_z)$ to model $q_\phi(z_{u_t}|u_t, s_{t:t+c}, \upsilon_{u_t})$ where $\mu_z$ and $\sigma_z$ are the mean and standard deviation outputted by the encoder. The decoder decodes $z_{u_t} \sim N(\mu_{z_{u_t}}, \sigma_{z_{u_t}})$ as following: $\hat{u}_t = g_{\psi_1} \circ p_{\psi_0}(z_{u_t}, s_t, \upsilon_{u_t})$, $g_{\psi_1}$ is a fully-connected layer for reconstruction. $\circ$ denotes cascade, i.e. the output of $p_{\psi_0}$ acts as the input of $g_{\psi_1}$. We utilize cascaded heads because traditional parallel heads, employed for both reconstruction and state residual prediction, can interfere with optimizing individual objectives and impede the learning of the shared representation [Azabou et al., 2021]. $p_{\psi_0}$ denotes the shared layers of the decoder. $\psi_{i \in \{1,2\}}$ denote the parameters of the prediction networks. The loss function of our VAE is:

$$L_{VAE}(\phi, \psi) = \mathbb{E}_{s,u \sim D, z \sim q_\psi}\left[\|\hat{u}_t - u_t\|_2^2 + D_{KL}\left(q_\phi(\cdot|u_t, s_{t:t+c}, \upsilon_{u_t})\|N(0, I)\right)\right], \tag{4}$$

where $D$ is the buffer. The first term is the reconstruction loss (using MSE) of the action sequence, the last term is the Kullback Leibler divergence $D_{KL}$ between the variational posterior of latent representation $z$ and the standard Gaussian prior. By using the reparameterization trick [Kingma and Welling, 2013], $\hat{u}_t$ is differentiable with respect to $\psi$ and $\phi$. For any latent variables $z_{u_t}$, it can be decoded into multi-step actions $\hat{u}_t$ conveniently by the VAE decoder. That is,

$$
\begin{aligned}
\textbf{Encoder} &: z_{u_t} \sim q_\phi(\cdot|u_t, s_{t:t+c}, \upsilon_{u_t}) \quad \forall s_{t:t+c}, u_t, \upsilon_{u_t} \\
\textbf{Decoder} &: \hat{u}_t = g_{\psi_1} \circ p_{\psi_0}(z_{u_t}, s_t, \upsilon_{u_t}) \quad \forall s, z_{u_t}, \upsilon_{u_t}
\end{aligned} \tag{5}
$$

## 4.2 Dynamic-Aware Multi-step Action Representation

In section 4.1, we introduce how to build a decodable representation space for multi-step actions. However, it is still inefficient to learn the policy and value functions in the latent action space learned by the above VAE. In comparison to the real-world action space, the current construction of our action space lacks a crucial attribute: dynamic semantic smoothness. This refers to the manifestation of differences in environmental impact through the Euclidean distance between points in the latent space, where closer points correspond to

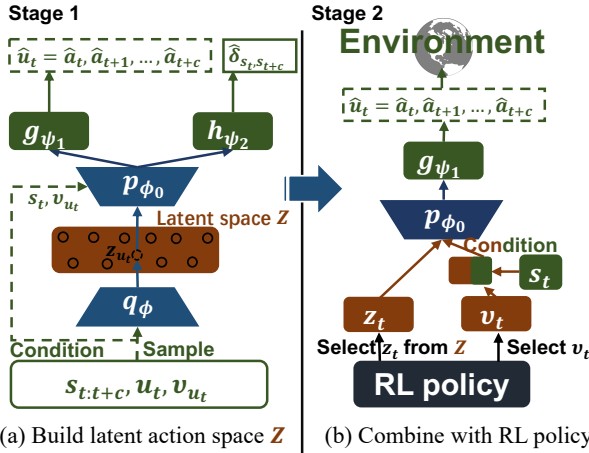

Figure 3: Architecture of MARS. 't' denotes different steps at different stage. Both $z_{u_t}$ and $z_t$ refer to variables in $Z$, but they are used in different stages (the same goes for $\upsilon_{u_t}$ and $\upsilon_t$).

more similar impacts. To address this issue, we further apply an unsupervised learning loss based on environmental dynamics prediction to refine the multi-step action representation. Relevant proof can be found in [Whitney et al., 2019]. MARS captures the environmental dynamics by predicting the state transition residual. We use the state transition as a measure of environmental dynamic becuase:
❶ State transition does not require a per-step reward, it can be used in reward-agnostic pretraining.
❷ Building value equivalent models is more difficult since Q-values in the early stage of training are inaccurate [Wang et al., 2016]. In contrast, state transition is more reliable and accessible. ❸ The same reward or Q-value may correspond to different environmental dynamics, but the same environmental dynamics have the same reward or Q-value.

Specifically, MARS predicts the residual difference between the state $s_{t+c}$ after the execution of $u_t$ and the current state $s_t$. As shown in the left of Figure 3, $h_{\psi_2}$ is a subnetwork of the decoder. For any transition sample $(s_t, u_t, s_{t+c})$, we denote the state residual as $\delta_{s_t, s_{t+1}} = s_{t+c} - s_t$ and denote $p_{state} = h_{\psi_2} \circ p_\psi$. The residual $\hat{\delta}_{s_t, s_{t+c}}$ is predicted as:

$$\hat{\delta}_{s_t, s_{t+c}} = p_{state}(z_{u_t}, s_t, \upsilon_{u_t}) \quad \forall\, s_t, z_{u_t}, \upsilon_{u_t}. \tag{6}$$

The environmental transition prediction loss is defined as:

$$L_{dy}(\phi, \psi) = \mathbb{E}_{s_t, u_t, s_{t+c}}\big[\|\hat{\delta}_{s_t, s_{t+c}} - \delta_{s_t, s_{t+c}}\|\big]. \tag{7}$$

Above all, the ultimate training loss for the multi-step action representation is:

$$L_{MARS}(\phi, \psi) = L_{VAE}(\phi, \psi) + \beta L_{dy}(\phi, \psi), \tag{8}$$

where $\beta$ is a hyper-parameter that controls the relative importance of the $L_{dy}$ and $L_{VAE}$. $L_{MARS}$ only depends on the environmental dynamic data which is reward-agnostic [Erraqabi et al., 2021, Yarats et al., 2021]. During training, transitions stored in the buffer or offline dataset are utilized. Additionally, we observe that MARS exhibits insensitivity to data quality. Notably, data collected through random policies suffice for effective training the multi-step action representation.

### 4.3 DRL with Multi-step Action Representation

As a plug-in method, MARS can be applied to any RL algorithm. It contains two types of actions: ❶ the encoded multi-step action $z$ and ❷ the action transition scale $\upsilon$. RL algorithms maximize the expected cumulative reward by selecting the optimal $z_t$ and $\upsilon_t$ at $s_t$. In this section, we apply MARS to a typical model-free RL method TD3 [Fujimoto et al., 2018] as an example. TD3 is a deterministic Actor-Critic algorithm. As illustrated in the right part of Figure 3, with the learned transition-aware multi-step action representation, the actor network learns a latent policy $\pi_\omega$ that outputs the latent actions according to current state $s$, i.e., $[z_t, \upsilon_t] = \pi_\omega(s)$. $z_t$ and $\upsilon_t$ respectively represent the action selected at time $t$ from $\mathcal{Z}$ and $\upsilon$ constructed in the section 4.1. Then we obtain the corresponding multi-step actions $u_t$ by decoding the latent action $z_t$ and $\upsilon_t$: $u_t = g_{\psi_1} \circ p_{\psi_0}(s, z_t, \upsilon_t)$.

---

**Algorithm 1** MARS-TD3

Initialize actor $\pi_\omega$ and critic networks $Q_{\theta_1}, Q_{\theta_2}$
Initialize conditional VAE $q_\phi, p_\psi$, buffer $D$.

**Stage 1: Build latent action space**

**while** not reaching warmup training times **do**
    Fill $D$ with data generated by random policy or offline datasets
    Update $q_\phi, p_\psi$ using samples in $D$. ▷ Eq.8
**end while**

**Stage 2: Train the RL policy**

**while** $t <$ policy training time **do**
    $z_t, \upsilon_t = \pi_\omega$ (with Gaussian noise)
    $u_t = g_{\psi_1} \circ p_{\psi_0}(z_t, s, \upsilon_t)$
    Execute $u_t$, observe $r$ and new state $s'$
    Fill $D$ with $(s, u_t, z_t, \upsilon_t, r, s')$
    update $Q_{\theta_1}, Q_{\theta_2}, \pi_\omega$    ▷ Eq.9, Eq.10
**end while**

---

Two critic networks $Q_{\theta_1}, Q_{\theta_2}$ are utilized which take the latent actions $z_t$ and $\upsilon_t$ as inputs to approximate the value function $Q_{\pi_\omega}$ more accurately. We train the critic network using the transition data $(s, \upsilon_t, z_t, r, s')$ sampled from the experience replay. To simplify notations, in this subsection, $s$ is the current state. $s'$ is the state perceived at the next interaction interval. The critic loss function is:

$$L_Q(\theta_i) = \mathbb{E}_{s, z_t, \upsilon_t, s'}\big[(y - Q_{\theta_i}(s, z_t, \upsilon_t))^2\big] \text{ for } \forall i \in \{1, 2\}. \tag{9}$$

Where $y = r + \gamma \min_{j=1,2} Q_{\bar{\theta}_j}(s', \pi_{\bar{\omega}}(s'))$, $\bar{\omega}$ denotes the target network parameters. The actor is updated according to the Deterministic Policy Gradient [Silver et al., 2014]:

$$\nabla_\omega J(\omega) = \mathbb{E}_s\big[\nabla_{\pi_\omega(s)} Q_{\theta_1}(s, \pi_\omega(s)) \nabla_\omega \pi_\omega(s)\big]. \tag{10}$$

The overall algorithm MARS-TD3 is summarized in Algorithm 1, which contains two major stages: ① the warmup stage and ② the policy learning stage. In stage 1, MARS is trained using a prepared replay buffer $D$. The SC-VAE is updated by minimizing the VAE and the environmental dynamic prediction loss. Note that the proposed algorithm has no requirement on how the buffer $D$ is prepared

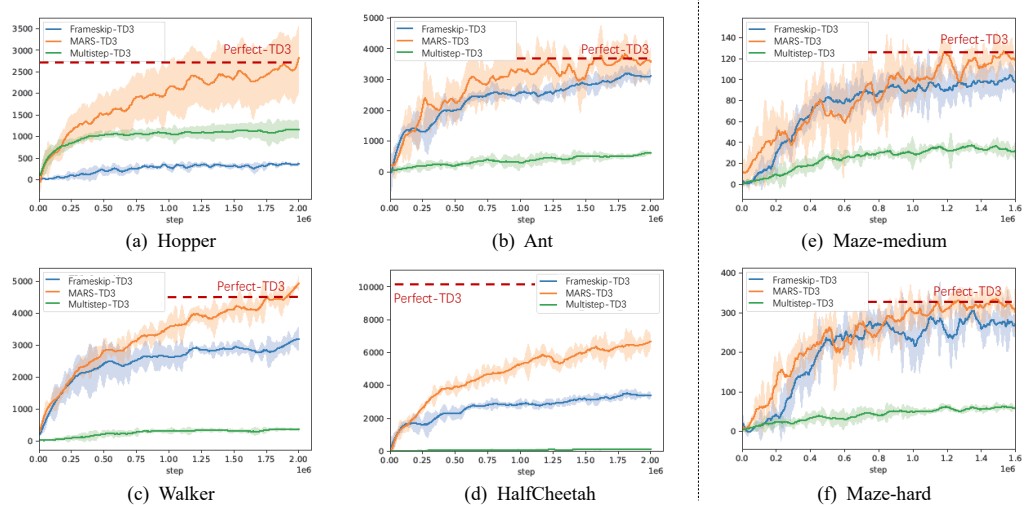

Figure 4: Comparisons of methods in simulated remote NPC control tasks with random interaction interval. The x- and y-axis denote the environment steps and average episode reward. Curves and shades denote the mean and the standard deviation over 8 runs.

and here we simply use a random policy to gather the data. In stage 2, given an environment state, the latent policy outputs the latent action $z_t$ and the action transition scale $v_t$ perturbed by the Gaussian exploration noise. Then, the latent action is decoded into the original multi-step actions by the decoder so as to interact with the environment. The newly collected transition sample is stored in the replay buffer $D$. After that, the latent policy is updated using the data sampled from $D$. The action representation model is also updated periodically in the second stage to make continual adjustments to the change of data distribution. The detailed network architectures are presented in appendix B.1.

**As for random interaction interval tasks,** the interaction interval cannot be predicted, so we let the agent output an action sequence with the maximum length. To improve the training stability, we precisely record the absolute time step and the execution flag of each action, which makes the actions be executed in strict accordance with the time step order. When a new action sequence arrives at the action executor, the previous action sequence will be replaced, and the execution flag of the unexecuted actions will be set to False. The subsequent rewards will be attributed to the actions executed in the new action sequence. Thus, the actual reward stored in the experience replay $D$ for each latent action is the sum of the executed action reward in the corresponding sequence.

## 5 Experiment

We empirically evaluate MARS to answer the following research questions. **RQ1: Performance in random interaction interval tasks.** Can MARS significantly improve the performance in random Intermittent-MDP tasks, such as simulated remote NPC control tasks? **RQ2: Performance in fixed interaction interval tasks.** Can MARS significantly improve the performance in fixed Intermittent-MDP problems, such as real-world robot arm grasping tasks? **RQ3: Generalization.** Can MARS be seamlessly integrated into existing RL algorithms and improve their performance? **RQ4: Ablation study.** Do both the action transition scale and the state dynamic prediction contribute to optimizing the multi-step latent action space? How is the robustness of MARS?

### 5.1 Random Intermittent-MDP Tasks (RQ1)

#### 5.1.1 Experimental Setups

**Benchmarks.** We select two types of control tasks to simulate the remote NPC control problem with random interaction intervals: (1) robot control and (2) navigation tasks. For robot control tasks, we select four typical openai Mujoco tasks with random interaction delays,i.e., Hopper, Ant, Walker, HalfCheetah. Mujoco is a well-known testbed and is widely used in reinforcement learning

research [Brockman et al., 2016]. For navigation tasks, we used the medium and difficult maps of 2dmaze in D4RL [Fu et al., 2020], where the agent's goal is to walk to the end of the maze under unstable interactions. Both types of tasks are modified to incorporate random interaction intervals, mimicking the real-world remote control scenarios [Chen and Wu, 2015]. In RTS games [Andersen et al., 2018], the maximum interaction interval usually spans $0.5s$ to $0.7s$, with the action execution time typically between $0.1s$ and $0.25s$. Accordingly, we set the maximum interval to 10 time steps, requiring the policy to generate an effective action sequence $a_{t:t+9}$ based on the received state $s_t$.

**Baselines.** To our knowledge, no specific solution exists for the Intermittent-MDP problem. Thus, we compare with three baseline methods. (1) **Perfect-TD3**: We train TD3 in the perfect environments with continuous interaction without interaction interval, i.e., normal MDP. It is the highest standard for evaluating MARS. (2) **Frameskip-TD3**: We combine the frameskip technique with TD3 (a common trick for multi-step decision) and apply it to the intermittent-MDP tasks. (3) **Multistep-TD3**: We modify TD3 to directly make decisions for the $c$ future steps by outputting a concatenated action vector of $c$ times of dimensionality.

### 5.1.2 Performance of remote NPC control tasks

To mitigate implementation bias and ensure a comprehensive comparison, we implement all methods using the same architecture based on TD3 [Fujimoto et al., 2018]. For all tasks, we set the dimension of $z_t$ to 8 and the scaling parameter $\beta$ to 5. We set the warm-up (stage 1) step to 400000 and 100000 for the Mujoco tasks and the navigation task respectively. Detailed parameter setting can be found in appendix B.2. The results in Figure 4 show that MARS-TD3 outperforms the other methods in all tasks, especially in the high-dimensional action control tasks (i.e., Mujoco). Compared with vanilla multstep-TD3, our method can significantly improve the performance of the RL algorithm in random Intermittent-MDP control problems by learning a more compact multistep action representation while avoiding the convergence difficulties caused by dimensional explosion. Note that MARS-TD3 can also achieve comparable performance with perfect-TD3 in most tasks.

## 5.2 Fixed Intermittent-MDP Tasks (RQ2)

### 5.2.1 Experimental Setups

**Task description.** The experiment involves a 6-DoF robot arm performing grasping tasks within a $30 \times 30 \times 30 cm^3$ tabletop workspace. 15 rounds of experiments are conducted using our method and baselines. In each round, 6 objects are randomly selected and placed on the table. The robot observes the workspace with a single-depth image from a fixed side view. The viewpoint of the virtual camera points toward the workspace's origin at a radial distance $r = 2l$ and an angle $\theta = \pi/3$, $l$ is the workspace's

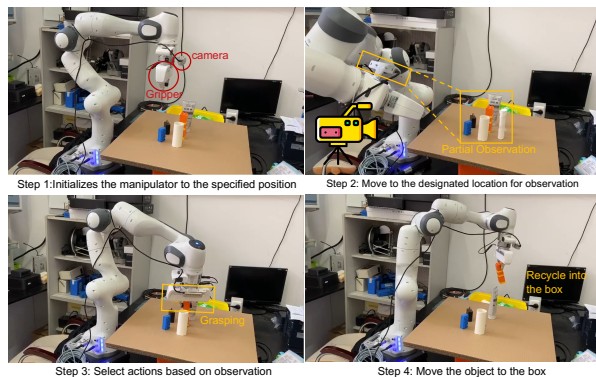

Figure 5: A complete grasp process in each interaction interval.

length. We follow the setting of traditional grasp tasks [Jiang et al., 2021] and set the total number of interactions to 6, meaning that the policy is expected to complete the grasp of all six targets in six intervals. Thus, the robot arm is allotted a single observation of the environment per object. As shown in Figure 5, at each interaction interval, there are 4 steps to grasp an object. The robotic arm has to first acquire observations at the beginning of each grasping. Upon perceiving the observation, an action sequence $a_{t:t+15}$ is executed to grasp the target object and then place the object in the target box. The time limit for grasping one object is set to $30s$. The robot need to move to a predefined location for capturing image (images are transformed into a $512$-dim vector). Action space is 3-dim[2], the reward function encourages policies to maximize the grasping success rate and motion smoothness.

**Baselines.** Online RL demands extensive exploration, resulting in low sample efficiency. Besides, Online RL accidental random exploration may lead to robot arm damage. Thus, we opt for offline

---

[2]The action denotes the coordinates expected to be reached by the hand, the defualt underlying planning algorithm generates the motion trajectory via the expected coordinates).

RL commonly utilized in robot scenarios, i.e. TD3+Behavior Cloning (TD3-BC) [Fujimoto and Gu, 2021], to serve as the backbone. Methods: (1) Vanilla TD3-BC. (2) Multistep TD3-BC, same as Multistep-TD3 used in the section 5.1 (3) Dense observation + TD3-BC. An extra camera provides dense observation every 3 seconds, and the robot arm still has to move to the observing position at the start of each episode to eliminate differences caused by the observing motion.

**Evaluation metrics.** (1) Grasp success rate (GSR), the ratio of successfully grasping the objects. (2) Declutter rate (DR), the average ratio of objects removed after successful grasping. (3) Motion smooth ratio (MSR): $\frac{\text{grasp time}}{\text{total time}}$, evaluating whether the motion is smooth and natural (pauses and shaking can result in lower scores). All results are averaged over 15 simulation rounds.

### 5.2.2 Performance of real-world Robot Arm Grasping

The overall comparison is presented in Table 1. The results show that due to the sparse interaction, the vanilla TD3-BC fails to complete the task and frequently pauses or takes unnatural motions during the execution. While the performance of TD3-BC is significantly improved with additional dense observations, it remains unsatisfactory due to the lack of real-time feedback. Moreover, the use of extra cameras results in

| Method | MSR(%) | GSR(%) | DR(%) |
|---|---|---|---|
| MARS + TD3-BC | 70.8 | 74.2 | 82.9 |
| Multistep + TD3-BC | 70.5 | 61.5 | 69.4 |
| Dense obs + TD3-BC | 65.3 | 59.1 | 70.7 |
| Vanilla TD3-BC | 22.5 | 34.1 | 27.3 |

Table 1: Performance of the robot arm grasping task.

higher costs of the experiment. Although Multistep + TD3-BC helps smoothen the motion, it fails to effectively mitigate the instability caused by Intermittent-MDP. MARS addresses the limitation of the vanilla Multistep + TD3-BC, which makes TD3-BC complete the task with high quality. We provide the corresponding motion pattern for each method. To further verify the effectiveness of MARS combined with online RL on fixed interaction interval control task. The results in the Appendix C.2 show that our method outperforms the baselines on all four tasks.

### 5.3 Generalization of MARS (RQ3)

We further test MARS with popular RL algorithms on three random interaction interval tasks: Hopper, Walker, and Maze hard. We maintain consistent parameters for each method and implement them based on the publicly available codebase. We train each RL algorithm with dense interactions as baselines and compare them with MARS-enhanced methods under the random interaction interval setting. Results in Appendix C.3 shows that all methods can learn effective policies with the help of MARS and the converged performance is comparable to the perfect interaction settings.

### 5.4 Ablation Study and Visual Analysis (RQ4)

We conducted evaluations on the key components of MARS, i.e. action transition scale and state dynamic prediction. Figure 6 (a) shows that both components optimize the latent space and improve the learning efficiency of DRL policies. MARS performs best when both modules are integrated. A comprehensive analysis is shown in Appendix C.4. Figure 6 (b) uses t-SNE [van der Maaten and Hinton, 2008] to visualize the latent action representations. We color each action based on its impact on the environment.

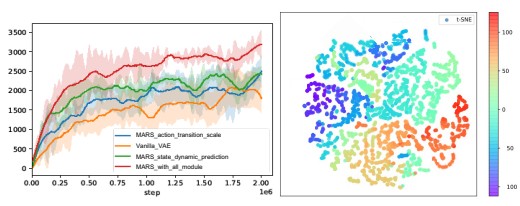

(a) Random Intermittent-MDP Hopper    (b) 2D t-SNE visualizations

Figure 6: Ablation study, the curve and shade denote the mean and a standard deviation of the returns over 5 runs.

Results show that actions with a similar impact, i.e., $\delta_{s_t, s_{t+c}}$, on the environment are clustered closely, which indicats that the learned action representation is dynamic smooth. Besides, results in Appendix C.5 improve the robustness of MARS to different interaction interval settings. Results in Appendix C.6 show the influence of latent space dimensions on MARS. Lastly, an analysis of self-supervised training steps can be found in Appendix C.7.

## 6 Conclusion and Limitation

In this paper, we observe that previous DRL methods fail to learn effective policies in intermittent control scenarios because of the discontinue interaction. To improve the performance of DRL on

such tasks, we propose **M**ulti-step **a**ction **r**epresentation (MARS) to construct a reliable multi-step latent action space. Based on this latent action space, DRL methods can make effective advance decisions to ensure the smoothness and efficiency of the agent's motion when it cannot interact with the environment. MARS outperforms baselines in a variety of fixed and random interaction intervals control tasks. Additionally, MARS has potential for improvement in represent extremely long action sequences, which we will address by identifying more powerful representation models, e.g. transformer based VAE, in the future.

## Acknowledgement

This work is supported by the National Natural Science Foundation of China (Grant Nos. 62422605, 92370132, 62106172), the National Key R&D Program of China (Grant No. 2022ZD0116402) and the Xiaomi Young Talents Program of Xiaomi Foundation.

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

# A  Preliminaries

**Markov Decision Process (MDP).** A standard MDP can be represented as a tuple: $(\mathcal{S}, \mathcal{A}, \mathcal{P}, \mathcal{R}, \gamma, T)$, where $\mathcal{S}$ denotes the state set, $\mathcal{A}$ denotes an action set, $\mathcal{P}$ is the transition function: $\mathcal{S} \times \mathcal{A} \times \mathcal{S} \to [0, 1]$ and $\mathcal{R}$ is the reward function: $\mathcal{S} \times \mathcal{A} \to \mathbb{R}$. $\gamma \in [0, 1)$ is a discount factor and $T$ is the decision horizon. The target of the agent is to optimize its policy to maximize the expected discounted cumulative reward.

**Frameskip.** Frame-skipping may be viewed as an instance of (partial) open-loop control, under which a predetermined sequence of (possibly different) actions is executed without heed to intermediate states. Aiming to minimize sensing, Kalyanakrishnan et al. [2021] proposes a framework for incorporating variable-length open-loop action sequences in regular (closed-loop) control. The primary challenge in general open-loop control is that the number of action sequences of some given length $d$ is exponential in $d$. Consequently, the main focus in the area is on policies to prune corresponding data structures [Braylan et al., 2015]. Since action repetition restricts itself to a set of actions with size linear in $d$, it allows for $d$ itself to be set much higher in practice. With frame-skipping, the agent is only allowed to sense every $d$ state: that is, if the agent has sensed a state $s_t$ at time step $t >= 0$, it is oblivious to states $s_{t+1}, s_{t+2}, ..., s_{t+d-1}$, and next only observes $s_{t+d}$.

**Variational Auto-encoder.** The variational auto-encoder (VAE) is a directed graphical model with certain types of latent variables, such as Gaussian latent variables. A generative process of the VAE is as follows: a set of latent variable $z$ is generated from the prior distribution $p_\theta(z)$ and the data x is generated by the generative distribution $p_\theta(x|z)$ conditioned on $z : z \sim p_\theta(z), x \sim p_\theta(x|z)$. In general, parameter estimation of directed graphical models is often challenging due to intractable posterior inference. However, the parameters of the VAE can be estimated efficiently in the stochastic gradient variational Bayes (SGVB) framework, where the variational lower bound of the log-likelihood is used as a surrogate objective function. In this framework, a proposal distribution $q_\theta(x|z)$, which is also known as a "recognition" model, is introduced to approximate the true posterior $p_\theta(x|z)$. The multilayer perceptrons (MLPs) are used to model the recognition and the generation models. Assuming Gaussian latent variables, the first term of Equation A can be marginalized, while the second term is not. Instead, the second term can be approximated by drawing samples $z^{(l)}(l = 1, ..., L)$ by the recognition distribution $q_\theta(x|z)$, and the empirical objective of the VAE with Gaussian latent variables is written as follows:

$$L_{VAE}(\phi, \psi) = \frac{1}{L} \sum_\theta (x|z^{(l)}) - KL\big(q_\phi(z|x)||N(0, I)\big) \tag{11}$$

# B  Experimental Details

## B.1  NETWORK STRUCTURE

| Layer | Actor Network | Critic Network |
|---|---|---|
| Fully Connected | (state dim, 256) | (statedim + $\upsilon$ dim + latent space dim, 128) |
| Activation | ReLU | ReLU |
| Fully Connected | (256, 128) | (256, 128) |
| Activation | ReLU | ReLU |
| Fully Connected | (128, latent space dim) and $\upsilon$ dim | (128, 1) |
| Activation | Tanh | None |

Table 2: Network Structures for DRL Methods

Our codes are implemented with Python 3.7.9 and Torch 1.7.1. All experiments were run on a single NVIDIA GeForce GTX 3090 GPU. Each single training trial ranges from 4 hours to 17 hours, depending on the algorithms and environments. We will open source code in the near future.

Our TD3 is implemented with reference to `github.com/sfujim/TD3` (TD3 source-code). DDPG and PPO are implemented with reference to `https://github.com/sweetice/Deep-reinforcement-learning-with-pytorch`. For a fair comparison, all the baseline methods have the same network structure (except for the specific components of each algorithm) as our MARS-TD3 implementation. As shown in Tab.2, we use a two-layer feed-forward neural network of

| Model Component | layer | dimension |
|---|---|---|
| | Fully Connected (encoding) | $(\mathbb{R}^x, 256)$ |
| | Fully Connected (condition) | (stae dim + $v$ dim, 256) |
| | Element-wise Product | ReLU (encoding), ReLU(condition) |
| | Fully Connected | (256, 256) |
| Conditional Encoder Network | Activation | ReLU |
| | Fully Connected (mean) | (256, latent space dim) |
| | Activation | None |
| | Fully Connected (log std) | (256, latent space dim) |
| | Activation | None |
| | Fully Connected (latent) | (latent space dim, 256) |
| | Fully Connected (condition) | (stae dim +$v$ dim, 256) |
| | Element-wise Product | ReLU (encoding), ReLU(condition) |
| | Fully Connected | (256, 256) |
| | Activation | ReLU |
| Conditional Decoder, Prediction Network | Fully Connected ($v$) | (256, action dynamic transition) |
| | Activation | None |
| | Fully Connected (reconstruction) | (256, multi-step action dim) |
| | Activation | None |
| | Fully Connected | (256, 256) |
| | Activation | ReLU |
| | Fully Connected (prediction) | (256, state dim) |
| | Activation | None |

Table 3: Network structures for the Multi-step action representation (MARS).

256 and 256 hidden units with ReLU activation (except for the output layer) for the actor network for all algorithms. For DDPG the critic denotes the Q-network. For PPO, the critic denotes the V-network. All algorithms (TD3, DDPG, PPO) output two heads at the last layer of the actor network, one for latent action and another for dynamic transition potential.

The structure of MARS is shown in Tab.3. We use element-wise product operation [Mahmood et al., 2018] and cascaded head structure [Fuchs et al., 2021] to our model[Ma et al., 2024, 2022].

## B.2  Hyperparameter

For all experiments, we use the raw state and reward from the environment, and no normalization or scaling is used. No regularization is used for the actor and the critic in all algorithms. An exploration noise sampled from N(0, 0.1) [Dong et al., 2009] is added to all baseline methods when selecting an action. The discounted factor is 0.99 and we use Adam Optimizer [Li et al., 2016] for all algorithms. Tab.4 shows the common hyperparameters of algorithms used in all our experiments.

| Hyperparameter | Frameskip-TD3 | Multistep-TD3 | MARS-PPO | MARS-TD3 | MARS-DDPG | |
|---|---|---|---|---|---|---|
| Actor Learning Rate | $1e^{-4}$ | $1e^{-4}$ | $1e^{-4}$ | $1e^{-4}$ | $1e^{-4}$ | $1e^{-4}$ |
| Critic Learning Rate | $1e^{-3}$ | $1e^{-3}$ | $1e^{-3}$ | $3e^{-4}$ | $3e^{-4}$ | $1e^{-3}$ |
| Representation Model Learning Rate | None | None | None | $1e^{-4}$ | $5e^{-3}$ | $5e^{-3}$ |
| Discount Factor | 0.99 | 0.99 | 0.99 | 0.99 | 0.99 | 0.99 |
| Batch Size | 128 | 128 | 128 | 128 | 128 | 128 |
| Buffer Size | $1e5$ | $1e5$ | $1e5$ | $1e5$ | $1e5$ | $1e5$ |

Table 4: A comparison of common hyperparameter choices of algorithms. We use 'None' to denote the 'not applicable' situation.

## B.3  Additional Implementation Details

For PPO, the actor network and the critic network are updated every 2 and 10 episode respectively for all environments. The clip range of the PPO algorithm is set to 0.2 and we use GAE [Sutton and Barto, 2018] for a stable policy gradient. For DDPG, the actor network and the critic network is updated at every 1 environment step. For TD3, the critic network is updated every 1 environment step and the actor network is updated every 2 environment steps.

The default latent action dim is 8, we set the KL weight in representation loss $L_{MARS}$ as 0.5. Environment dynamic prediction loss weight $\beta$ is 5 (default).

# C  Additional experiment

## C.1  Performance of model-based reinforcement learning algorithms on Intermittent-MDP tasks.

| Methods | Ant (fixed-Intermittent) | Hopper (fixed-Intermittent) | Ant (random-Intermittent) | Hopper (random-Intermittent) |
|---|---|---|---|---|
| TD-MPC | $1795.4 \pm 375.6$ | $1795.4 \pm 214.8$ | $1447.2 \pm 694.8$ | $1073.6 \pm 157.1$ |
| Dreamer-v2 | $1648.2 \pm 417.5$ | $788.1 \pm 116.4$ | $1064.7 \pm 694.8$ | $974.7 \pm 201.8$ |
| TD3-multistep | $2673.6 \pm 316.8$ | $1359.7 \pm 258.3$ | $2795.4 \pm 264.1$ | $1211.6 \pm 169.5$ |
| MARS-TD3 | $2572.9 \pm 248.1$ | $3762.7 \pm 371.4$ | $3105.7 \pm 412.6$ | $2647.9 \pm 204.8$ |

Table 5: Comparison between MBRL and MFRL in intermittent control tasks, average of 3 runs.

The results in Table 5 show that the model-based reinforcement learning approach is significantly lower than our approach in all four scenarios, and even slightly worse than using TD3 for direct multi-step decision making. We analyze that this is because the errors caused by the mismatch (sub-optimal) of the dynamic model will accumulate due to multi-step decision-making, resulting in sub-optimal policy. Unfortunately, the training of dynamic models is data hungry (high cost), that is, a large amount of high-quality expert data is required to ensure the accuracy of the model shop, which is difficult to obtain, especially in real-world scenarios.

## C.2  Validation of the combination of MARS and online methods

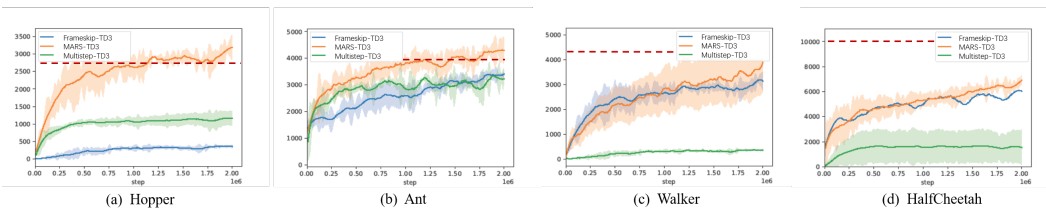

Figure 7: The performance of the methods on four simulated tasks. The curve and shade denote the mean and a standard deviation over 5 runs.

We use the mainstream online reinforcement learning algorithm TD3 in combination with MARS and compare it with the baseline mentioned in Sec.5.1 in four tasks. We set the interval to 10 time steps, requiring the policy to generate an effective action sequence $a_{t:t+9}$ based on the received state $s_t$.

For all tasks, we set the dimension of $z_t$ to 12 and the scaling parameter $\beta$ to 4. We set the warm-up (stage 1) step to 300000 and 100000 for the Mujoco tasks and the navigation task respectively. The results in Figure 7 show that MARS-TD3 outperforms the other baselines in all fixed Intermittent-MDP tasks and achieve comparable performance with perfect-TD3 in most tasks. **This further proves that MARS can effectively improve the effectiveness of Online DRL on fixed Intermi tasks.**

## C.3  Generalization of MARS

We test MARS with popular RL methods on three tasks: Hopper, Walker, and hardMaze. To make the experiment fair, we used the same parameters for all methods and implemented them based on public code. We use each RL algorithm to train on three tasks under the ideal setting and compare them with their corresponding improvement methods. To show the optimal score after the algorithm convergence, we train all the algorithm's 2000000 time steps. The results in Tab.6 show that all methods can learn effective policies with the help of MARS and perform similarly to their ideal settings. The differences in scores are mainly due to the variation in performance of the RL algorithms. In summary, MARS can be combined with different methods to provide a reliable action space for solving Intermittent-MDP as normal MDP with RL.

## C.4  Details of Ablation study

| Benchmarks | MARS-PPO | MARS-DDPG | MARS-TD3 |
|---|---|---|---|
| Maze hard | 256 \| 0.7 ↑ | 243 \| 2.5 ↑ | 311 \| 16.3 ↑ |
| Hopper | 2851.4 \| 13.5 ↓ | 1815.6 \| 184.3 ↑ | 3384 \| 53.1 ↑ |
| Walker | 3831.2 \| 285.1 ↓ | 1032.7 \| 201.9 ↓ | 4821.6 \| 427.6 ↑ |

Table 6: The parameters of all methods are optimized by grid search. The results of applying MARS to popular RL algorithms on three random interaction interval tasks. The maximum interaction interval is set to 8. Each data in the table is in the following format: MARS-RL score | the score difference compared to the perfect dense interaction baseline. ↓ denotes the score of MARS lower than the dense interaction baseline. ↑ denotes the score of MARS is higher. All scores are averaged over 5 runs.

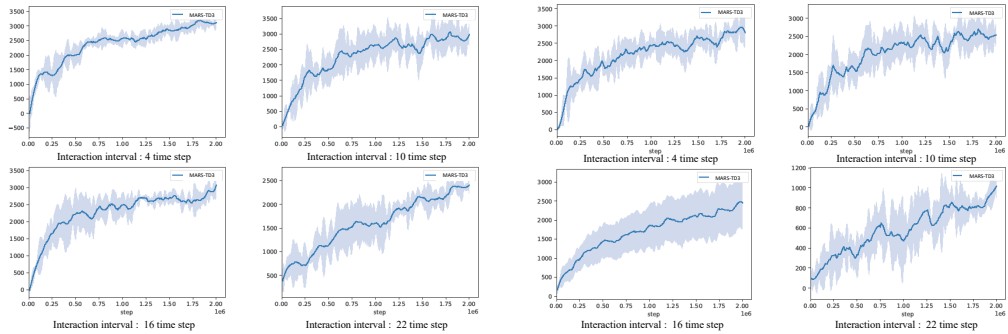

(a) Fixed Intermittent-MDP scenarios      (b) Random Intermittent-MDP scenarios

Figure 9: The curve and shade denote the mean and a standard deviation over 5 runs.

We conducted two experiments to show how well the two mechanisms of MARS work together. Although the results of randomized Intermittent-MDP and fixed Intermittent-MDP are slightly different, the same conclusion can be derived: The green curves in Figure 8 demonstrate that the representation model

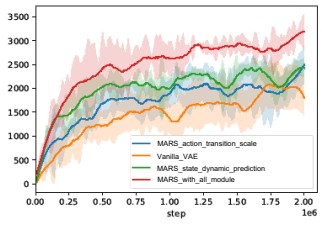 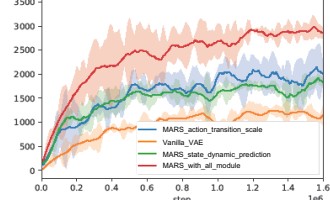

(a) Fix Intermittent-MDP Walker      (b) Random Intermittent-MDP Walker

Figure 8: Details of ablation study. The curve and shade denote the mean and a standard deviation over 5 runs.

with increased action transition scale is much better than the original VAE. This means that dynamic transition potential can create an latent action space by explicitly modeling the dependence between multi-step actions. The blue curves also show that VAE with state dynamic prediction is better than the original VAE because it can represent action sequences that have similar environmental effects at close locations. Finally, the red curves show that the two mechanisms work well together in MARS, and combining them improves representation ability.

## C.5 Validity verification of multi-style interaction intervals

To further demonstrate the effectiveness of MARS in diverse intermittent control scenarios. For fixed interaction control tasks, we uniformly set the forbidden interaction duration and conducted four experiments on Hopper. The results in Figure 9(a) show that MARS can solve most tasks effectively and still guarantee good scores at long intervals, but the effectiveness of MARS decreases significantly when the interval is too long (which is not common in real-world scenarios). We believe that this is because VAE is unable to effectively characterize excessively long sequences, leading to the failure of multi-step action space modeling.

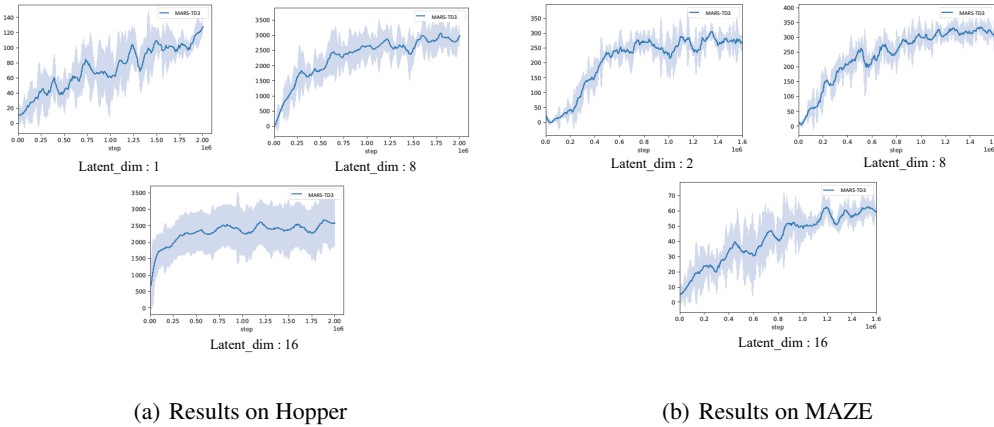

(a) Results on Hopper                    (b) Results on MAZE

Figure 10: The curve and shade denote the mean and a standard deviation over 5 runs.

In addition, to observe the sensitivity of MARS to interaction intervals on random Intermittent-MDP tasks, we uniformly set the forbidden interaction duration and conducted four experiments on Hopper. The results in Figure 9(b) show that in random Intermittent-MDP scenarios, MARS performs well in both short and medium-interval scenarios. However, convergence changes slowly in the very long interval scenario, and the score is only half that of the medium interval task. Because MARS's representational capabilities are not perfect for modeling long action sequences for extremely long-spaced tasks (even if this setting rarely occurs in real-world scenarios). Therefore, in the future, we hope to find more suitable representation models to overcome this problem.

## C.6 The influence of Latent action space dimension on algorithm effect

The representation space dimension of VAE is an important hyperparameter. If the latent space dimension is too low, a large amount of original data information will be lost, resulting in invalid representation space. On the contrary, when the latent space dimension is too large, the calculation amount of the model will be increased. To verify the sensitivity of MARS to latent space dimensions, we test it on two tasks with different original action dimensions. We set up four sets of latent space dimensions for fixed Intermittent-MDP Hopper (interaction interval time step: 8, original action dimension: 3, so the action sequence dimension to be modeled is 24). The learning curve in Figure 10(a) shows that for raw data of such high dimensions, when the latent space dimension is set too low, the latent space information will be lost, resulting in the convergence failure of reinforcement learning policies. On the contrary, too high a latent space dimension increases the complexity of reinforcement learning policy exploration.

In addition, we set up four comparison experiments on the 2dmaze task with a lower dimension of the original action sequence (interaction interval time step: 4, original action dimension: 2, so the action sequence dimension to be modeled is 8). The experimental results in Figure 10(b) show that the suboptimal policy can be learned when the latent space dimension is low, because the original data dimension is low. So the low-dimensional latent space loses less information. The score increases as the latent space dimension increases. However, when the latent space dimension is too high, the score will drop significantly, which is because of the exploration difficulties brought by high-dimensional latent space.

## C.7 The influence of environment steps of warmup stage

In this section, we conduct some additional experimental results for a further study of MARS from different perspectives: We provide the exact number of samples used in the warm-up stage (i.e., stage 1 in Algorithm 1 in each environment in Tab.7. The number of warm-up environment steps is about $5\% \sim 10\%$ of the total environment steps in our original experiments. Moreover, we also conducted some experiments to further reduce the number of samples used in the warm-up stage (at

most $80\%$ off). See the colored results in Tab.7. MARS can achieve comparable performance with $< 3\%$ samples of the total environment steps.

Conclusion: The number of warm-up environment steps is about $5\% \sim 10\%$ of the total environment steps in our original experiments. The number of warmup environment steps can be further reduced by at most $80\%$ off (thus leading to $< 3\%$ of the total environment steps) while the comparable performance of our algorithm remains.

| Environment | Warm-up steps (original) | Warm-up steps (new) | Total Env. Steps |
|---|---|---|---|
| Hopper | 400000(0.08\|3219.1) | 100000(0.02\|3086.4) | 5000000 |
| Ant | 400000(0.08\|4305.7) | 100000(0.02\|4025.6) | 5000000 |
| Walker | 400000(0.08\|4961.3) | 100000(0.02\|4792.6) | 5000000 |
| HalfCheetah | 400000(0.08\|6593.2) | 100000(0.02\|6071.2) | 5000000 |
| 2dmaze-medium | 100000(0.083\|127.8) | 30000(0.025\|118.5) | 1200000 |
| 2dmaze-hard | 100000(0.083\|327.6) | 35000(0.0292\|296.1) | 1200000 |

Table 7: The exact number of samples used in warm-up stage training in different environments. The column of 'original' denotes what is done in our experiments; the column of 'new' denotes additional experiments we conduct with fewer warm-up samples (and proportionally fewer warm-up training). For each entry $x(y|z)$, x is the number of samples (environment steps), y denotes the percentage number of $\frac{warm-up\ environment\ steps}{number\ of\ total\ environment\ steps\ during\ the\ training\ process}$, and z denotes the corresponding performance of MARS-TD3.

