# OpenReview forum: "Unlock the Intermittent Control Ability of Model Free Reinforcement Learning"
_NeurIPS.cc/2024/Conference — NeurIPS 2024 poster_

### Official Review · Reviewer_Bn1i · 2024-07-11

**Soundness:** 3
**Presentation:** 3
**Contribution:** 3
**Rating:** 5
**Confidence:** 4

**Summary:**

This paper proposes a new method for RL problems that needs to propose a sequence of actions at each step due to latency of the environment. The method is straightforward in that it uses VAE to encode the action space of a consecutive of actions. Empirical results show that this simple technique improves the performance and sample efficiency.

**Strengths:**

1. The paper is well-written and easy to follow.
2. The method is simple but the performance improvement is significant.

**Weaknesses:**

The setting looks like the delay MDPs but is simpler as the observation is not delayed. The paper does not compare with methods for delay MDPs. The method is straightforward and easy to think of.

**Questions:**

1. Why define intermitten MDP separately while delayed MDP is widely adopted in the community?
2. How does the method compare to existing delay MDP methods?

**Limitations:**

See weakness.

---

> ### Author Rebuttal · Authors · 2024-08-06
>
> We are grateful to you for recognizing the importance of our research. Your suggestions inspired us to improve our work, we analyzed the difference between our setting (intermittent MDP) and Delayed-MDP through demo examples and detailed explanations in the revised version. Additionally, we compared our method with several SOTA delayed MDP methods on multiple complex tasks in the main experiment section (4 more robotic motion control tasks and 2 more manipulation tasks).
>
> If you think the following response addresses your concerns, we would appreciate it if you could kindly consider raising the score.
> ## Part 1
> ### Questions
> **Q1: Compare Intermitted MDP and Delayed MDP: Why define intermittent MDP separately while delayed MDP is widely adopted in the community?**
>
> We described our setting more clearly in the new version. Both the Intermittent MDP and the traditional Delayed MDP models aim to represent non-ideal (unstable) environments. However, the tasks differences that both focus on also make the following distinction：
>
> **Reward feedback:** Mostly, in delayed MDP, the reward associated with each state-action pair can be acquired independently (**dense reward**)[1]. However, Intermittent MDP regards reward feedback as the cumulative rewards of a sequence of actions (**sparse reward**), a depiction that aligns more realistically with scenarios in practice.
> - We compare the performance of the delayed MDP methods with our method on Ant-v2 when dense rewards are not available (simulate lost state by repeating the last reward in the sequence). The results show that our method is more effective in dealing with fuzzy reward feedback. We use two recent SOTA delayed MDP methods, i.e. DCAC[2] and State Augmentation(SA)[3], as representatives of this class of methods. The setup is the same as that in the main experiments.
>
> Table 1. Performance (Average of 3 runs):
> | Method      | Intermittent Control Ant-v2|
> | :-: | :-: |
> | Ours |$4354.61\pm 158.44$|
> | DCAC  |$3281.93\pm 544.17$|
> | SA |$3446.21\pm 262.59$|
>
> **Motion mode requirements (described in Fig.1):** The real-world tasks that Intermittent MDP targets typically always involve high-frequency operational demands, e.g. game NPC control and robot control[4]. Thus, **Intermittent MDP expects the action to be executed at every time step to guarantee smooth and stable motion. In contrast, delayed MDP does not incorporate this constraint and permits certain time steps to be non-execution[3].** For instance, in a scenario where a robot needs to continue walking despite a blocked interaction channel, it cannot afford to wait for the next state to be successfully transmitted before taking action. Delaying the decision in such cases could result in the movement abruptly halting within a specific timeframe, leading to a loss of balance and a potential fall. So the key in the above task is not solely on the efficiency of some particular executing actions. Rather, the emphasis lies on ensuring that each action smoothly transitions with its neighboring actions while maintaining validity.
> - We compared the motion smoothness between our method and delayed MDP methods in the Humanoid scenario,  Humanoid requires smooth fine-tuning at each time step to maintain the balance of the torso effectively. We use the Smoothness rate to measure the motion (whether the motion is coherent and stable, invalid jitter and stagnation will reduce the score, please refer to Sec.4.2 for details). The results show that MARS can make the movement of the executor smoother and more stable.
>
> Table 2. Performance (smoothness rate (%)) (Average of 3 runs):
> | Method| Intermittent Control Ant-v2|
> | :-: | :-: |
> | Ours |$4354.61\pm 158.44$ (78)|
> | DCAC  |$3495.68\pm 244.51$ (42)|
> | SA |$3108.29\pm 194.35$ (51)|
>
> **The information used for decisions:** Delayed MDP methods typically involve accessing prior information, such as the time to be delayed or even intermediate states, to enhance decision-making [2]. Our constraint is more stringent, where the agent is only permitted to make advanced decisions based on the current state.
> - We compare the effect of the delayed MDP method without auxiliary information (simulate lost state by using zero mask) and our method, on the Intermitted control task. The results show that our method is somewhat more robust to sparse information.
>
> Table 3. Performance (Average of 3 runs):
> | Method| Intermittent Control Ant-v2|
> | :-: | :-: |
> | Ours |$4354.61\pm 158.44$|
> | DCAC  |$2795.68\pm 485.83$|
> | SA |$3371.46\pm 169.27$|
>
> **Decision step number per step:** The delay scenario addressed by delayed MDP involves brief durations of delay (statistics from [2] indicate that the majority of delays are less than one second), making the future decision action length relatively short in this context. In contrast, our setup necessitates accounting for instances where the communication channel may remain non-functional for an extended time due to channel breakdowns. Therefore, Intermittent MDP method must consider longer durations (in real scenarios, there could be intervals exceeding 5 seconds [4]) in our deliberations, i.e. Deciding on a lengthy action sequence in a single step.
> - We tested the performance with various single-step decision action sequence lengths for the corresponding methods of the two MDPS in the Walker2d scenario (DCAC is the easiest method to change to a multi-step decision mode in delayed MDP). Tab. 4 shows that MARS performs better in the long decision sequence setting.
>
> Table 4. Performance (Average of 3 runs):
> | Method| Decision step $c=4$|$c=8$|$c=12$|$c=16$|
> | :-: | :-: |:-: |:-: |:-: |
> | MARS|$5309\pm 143.85$|$5283\pm 171.46$|$5309\pm 143.85$|$5194\pm 201.52$|
> | DCAC|$5134\pm 118.64$|$4796\pm 208.72$|$2962\pm 571.07$|$2836\pm 485.11$|
>
> **Due to the word limit, the reference and second part of the response are in the following comment. We are grateful for your thorough and conscientious reviewing.**

---

> ### Author Response · Authors · 2024-08-07
> **Rebuttal by Authors**
>
> ## Part 2
> **Q2: How does the method compare to existing delay MDP methods?**
>
> Thanks for your suggestion. we add a comparison of our method with mainstream delayed MDP methods on multiple complex tasks from the perspective of multiple metrics in the new version.
>
> *Baselines:* We select two recent SOTA methods in the Delayed MDP domain, i.e. DCAC [2] and State Augmentation (SA) [3] (Relax the Intermitted setting restrictions, allow such methods to additionally use dense rewards and delay priors at each step, and set the delay coefficient to the interaction interval). For a more comprehensive analysis, a recent model-based approach,i.e. delayed Dreamer[5], is also chosen.
>
> *Benchmarks:* We further select four more challenging DeepMind Control tasks focused on bionic robot locomotion: Dog Run, Dog Trot, Dog Stand and Humanoid Walk. DMC tasks demand coordinated movement from multi-joint bionic robots. Besides, two robotic arm control tasks in MetaWorld: Sweep Into and Coffee Push are used. The environmental parameters and network hyperparameters remained consistent with the main experiment.
>
> *Metrics:* We evaluate methods in terms of performance and Smoothness (whether the motion is coherent and stable, invalid jitter and stagnation will reduce the score, please refer to Sec.5.2 for details).
>
> Table 5. Performance in newly added difficult tasks (smoothness (%)) (Average of 4 runs):
> | Method      |Dog Run|Dog Trot| Dog Stand| Humanoid Walk|Sweep Into|Coffee Push|
> | :-----------: | :-----------: | :------------: | :-----------: | :-----------: | :-----------: | :-----------: |
> | **Ours**|$124.61\pm 44.92 (75)$|$574.12\pm 28.76 (88)$|$614.03\pm 17.42 (73)$|$105.47\pm 35.83 (82)$|$0.51\pm 0.13 (68)$|$0.38\pm 0.06(63)$|
> | DCAC|$96.87\pm 28.44 (53)$|$426.93\pm 50.48 (72)$|$562.64\pm 22.73 (64)$|$105.32\pm 29.16 (49)$|$0.44\pm 0.27 (41)$|$0.19\pm 0.13 (48)$|
> | SA |$92.74\pm 51.06 (37)$|$385.67\pm 52.49 (39)$|$503.94\pm 14.86 (27)$|$75.66\pm 31.42 (45)$|$0.52\pm 0.21 (37)$|$0.34\pm 0.09 (39)$|
> | delayed Dreamer |$95.31\pm 26.74 (46)$|$428.39\pm 46.23 (46)$|$526.07\pm 21.84 (25)$|$89.25\pm 27.41 (41)$|$0.56\pm 0.17 (32)$|$0.39\pm 0.04 (37)$|
>
> Table 6. Performance in old tasks (smoothness (%)) (Average of 4 runs):
> | Method  |Ant-v2|Walker2d-v2| HalfCHeetah-v2| Hopper-v2|
> | :-: | :-: | :-: | :-: | :-: |
> | **Ours**|$4354.61\pm 158.44$ (78)|$5436.42\pm 217.83$ (82)|$6175.63\pm 273.95$ (76)|$2613.58\pm 177.96$ (83)|
> | DCAC |$3279.82\pm 127.83$ (42)|$4892.18\pm 383.07$ (54)|$5811.51\pm 108.33$ (58)|$2684 .31\pm 238.27$ (63)|
> | SA |$3492.53\pm 131.95$ (51)|$2584.18\pm 106.24$ (59)|$3281.45\pm 139.42$ (66)|$381.74\pm 53.67$ (71)|
> | delayed Dreamer|$2408.25\pm 31.76$ (35)|$1529.26\pm 68.21$ (63)|$112.73\pm 17.82$ (68)|$1173.93\pm 78.28$ (78)|
>
> Table 5,6 shows that our method performs better than Delayed MDP methods in almost all intermitted MDP control tasks while ensuring smooth and coherent motion of the agent.
>
> *P.S. If you know other delayed MDP methods that you would like us to compare but are not in the scope of our investigation, please let us know and we would be happy to include them in our experimental analysis.*
>
> [1]: Du, Keliang, et al. "Random-Delay-Corrected Deep Reinforcement Learning Framework for Real-World Online Closed-Loop Network Automation." Applied Sciences 2022.
>
> [2]: Ramstedt, Simon, et al. "Reinforcement learning with random delays." ICLR 2020.
>
> [3]: Nath, Somjit, et al. "Revisiting state augmentation methods for reinforcement learning with stochastic delays." CIKM 2021.
>
> [4]: Jiang, Zhenyu, et al. "Synergies between affordance and geometry: 6-dof grasp detection via implicit representations." Transactions on Robotics 2021.
>
> [5]: Karamzade A, Kim K, Kalsi M, et al. "Reinforcement learning from delayed observations via world models." PMLR 2024.
>
> **The Third part of the response is in the following comment. We are grateful for your thorough and conscientious review.**

---

> ### Author Response · Authors · 2024-08-07
> **Rebuttal by Authors**
>
> ## Part 3
> ### Weaknesses
> **W1: The method is straightforward and easy to think of.**
>
> We add a description of the novelty of our work. The Backbone of MARS is straightforward, a C-VAE, thereby enhancing convenience and plug-and-play capabilities from an application standpoint. However, at the practice level, we discovered that the original C-VAE could not effectively construct the semantic smooth latent space and the representation realization of long action sequences was poor, thereby constraining the optimization of the RL algorithm. Thus, the primary innovation of our method revolves around enhancing the capability of constructing the latent space of the original VAE through the development of more effective auxiliary techniques.
>
> In particular, we introduce two innovative techniques for the original VAE: 1) the introduction of action transition scale (ATS) to dynamically restrict action decoding within an effective interval to ensure policy effectiveness (Sec 4.1), and 2) the incorporation of a state residual module (SR) to encourage points in the latent space with similar environmental impacts to be closer to each other (Sec 4.2), enhancing the overall model performance.
>
> Experiments are conducted on Ant-v2 to assess the efficacy of the two key modules. The results in Tab. 7 indicate that RL optimization is more efficient with these modules. Additionally, Table 8 shows that the performance of MARS surpasses that of the original VAE in long-sequence decision scenarios.
>
>
> Table 7. Performance (smoothness (%)) (Average of 3 runs):
> | Method  |Ant-v2|
> | :-: | :-: |
> | VAE + ATS + SR (MARS) |$4354.61\pm 158.44$|
> | VAE + SR |$4016.47\pm 213.06$|
> | VAE + ATS |$4122.18\pm 65.37$|
> | vanilla VAE|$3685.92\pm 362.72$|
>
> Table 8. Performance in Walker2d  (Average of 4 runs):
> | Method| Decision step $c=4$|$c=8$|$c=12$|$c=16$|
> | :-: | :-: |:-: |:-: |:-: |
> | VAE + ATS + SR (MARS) |$5309\pm 143.85$|$5283\pm 171.46$|$5309\pm 143.85$|$5194\pm 201.52$|
> | vanilla VAE |$4623.51\pm 463.81$|$3941.21\pm 297.44$|$3806.14\pm 538.24$|$3612 .23\pm 635.42$|

---

> ### Author Response · Authors · 2024-08-11
>
> Dear reviewer Bn1i, we sincerely apologize for the inconvenience caused by placing our important experimental results in the general PDF, which may have required additional time for you to locate them. To address this, We have strategically placed the experiment that should be added to the PDF onto the webpage (integrated with responses to each question) to enhance the readability.
>
> If you have any further questions or suggestions, please feel free to share them with us. Your feedback is invaluable and catalyzes enhancing our work.

---

> > ### Comment · Reviewer_Bn1i · 2024-08-14
> > **Score Raised**
> >
> > Thanks for the detailed explanation and comparison. I suggest to add these experiments in the main paper in the next version.

---

> ### Author Response · Authors · 2024-08-12
> **Supplement for comparison experiments between our method and the delayed MDP methods**
>
> We enhanced the comparison experiment between the MARS and delayed MDP methods by increasing the number of seeds from 4 to 8. Furthermore, two new baselines are added to enhance the richness of the experiment.  The experimental results show that the advantage of our method is further improved as the number of seeds doubles.
>
> *Baselines:* We append two newest baselines to the three existing SOTA delayed MDP methods (DCAC, SA, delayed Dreamer): BPQL [1], the latest Actor-critic-based continuous control algorithm for delayed feedback environments. AD-RL [2],  a SOTA method that utilizes auxiliary tasks with short delays to accelerate RL with long delays.
>
> *Benchmarks:* We chose six difficult tasks. DeepMind Control tasks: Dog Run, Dog Trot, Dog Stand and Humanoid Walk. Besides, two robotic arm control tasks in MetaWorld: Sweep Into and Coffee Push are used. The environmental parameters and network hyperparameters remained consistent with the main experiment.
>
> *Metrics:* We evaluate methods in terms of performance and Smoothness (whether the motion is coherent and stable, invalid jitter and stagnation will reduce the score, please refer to Sec.5.2 for details).
>
> Table 1. Performance in newly added difficult tasks (smoothness (%)):
> | Method      |Dog Run|Dog Trot| Dog Stand| Humanoid Walk|Sweep Into|Coffee Push|
> | :-----------: | :-----------: | :------------: | :-----------: | :-----------: | :-----------: | :-----------: |
> | **Ours**|$124.61\pm 24.71 (78)$|$592.42\pm 28.76 (88)$|$626.11\pm 18.36 (76)$|$120.82\pm 26.45 (85)$|$0.53\pm 0.04 (74)$|$0.42\pm 0.05(69)$|
> | DCAC|$91.48\pm 16.75 (55)$|$411.06\pm 36.52 (76)$|$538.47\pm 22.73 (64)$|$101.63\pm 15.28 (46)$|$0.42\pm 0.08 (48)$|$0.23\pm 0.11 (51)$|
> | BPQL|$95.37\pm 20.31 (62)$|$451.72\pm 36.67 (78)$|$526.13\pm 17.92 (66)$|$92.84\pm 22.51 (61)$|$0.47\pm 0.11 (53)$|$0.27\pm 0.05 (57)$|
> | AD-RL|$88.26\pm 14.03 (64)$|$448.49\pm 24.72 (72)$|$471.82\pm 21.17 (51)$|$105.32\pm 15.25 (57)$|$0.39\pm 0.08 (45)$|$0.23\pm 0.07 (41)$|
> | SA |$93.26\pm 28.14 (33)$|$384.26\pm 45.03 (42)$|$486.91\pm 10.51 (32)$|$78.21\pm 27.41 (47)$|$0.49\pm 0.07 (42)$|$0.33\pm 0.06 (34)$|
> | delayed Dreamer |$95.28\pm 19.42 (46)$|$416.65\pm 24.18 (45)$|$526.07\pm 11.52 (38)$|$92.07\pm 13.59 (43)$|$0.48\pm 0.08 (36)$|$0.36\pm 0.03 (46)$|
>
> Table 1 shows that our method performs better than Delayed MDP methods in almost all intermitted MDP control tasks while ensuring the smooth and coherent motion of the agent.  Besides, the advantage of our method is further improved as the number of seeds doubles.
>
> [1]: Kim, Jangwon, et al. "Belief projection-based reinforcement learning for environments with delayed feedback."NuerIPS 2023.
>
> [2]: Wu, Qingyuan, et al. "Boosting Long-Delayed Reinforcement Learning with Auxiliary Short-Delayed Task." ICML 2024.

---

### Official Review · Reviewer_5oZS · 2024-07-12

**Soundness:** 3
**Presentation:** 2
**Contribution:** 3
**Rating:** 6
**Confidence:** 4

**Summary:**

The paper introduces Multi-step Action RepreSentation (MARS) to address intermittent control problems in reinforcement learning. Intermittent control refers to situations where the interaction between the decision maker and the executor is discontinuous due to interruptions or communication issues. MARS encodes a sequence of actions into a compact and decodable latent space, allowing RL algorithms to optimize and learn smooth and efficient motion policies. Experiments are conducted on both simulated and real-world tasks, demonstrating that MARS significantly improves learning efficiency and performance compared to existing baselines.

**Strengths:**

* The paper addresses an interesting problem of intermittent control in RL, which has not been broadly studied in the RL community.
* The concepts of action sequence representation and action transition scale in MARS sound interesting and effective to me.
* The paper provides a thorough explanation of the MARS method, including the encoding and decoding process, the use of action transition scale, and the state dynamic prediction. The experiments include both simulation tasks and real-world robotic grasping tasks and demonstrate the effectiveness of MARS in improving learning efficiency and performance.
* The paper is easy to follow.

**Weaknesses:**

* The comparisons in the experiment are with some simple and intuitive baselines. While the authors mention that no specific solution exists for this problem, it would still be helpful to discuss related work in other research fields.
* The experiments include both simulated and real-world tasks, but overall the number of tasks is a bit limited. It would be beneficial to include more tasks to validate the effectiveness of MARS.

Minors:
* Figure 6 is not adequately described, e.g., what the task is and what Vanilla_VAE stands for.
* Stage 2 in Algorithm 1 should be revised according to Line 276: The action representation model is also updated periodically in the second stage to make continual adjustments to the change of data distribution.
* Eq. (3): $\sum_{i=t}^{c-1} \rightarrow \sum_{i=t}^{c+t-1}$.
* Line 230: $\delta_{s_t,s_{t+1}} \rightarrow \delta_{s_t,s_{t+c}}$
* Line 231: $p_{state}=h_{\psi_2}\circ p_\psi \rightarrow p_{state}=h_{\psi_2}\circ p_{\psi_0}$
* Line 222: becuase $\rightarrow$ because

**Questions:**

* Why does the input of the encoder $q_\phi$ include $s_{t:t+c}$? According to the description of the intermittent control task (Line 31: agents are unable to acquire the state sent by the executor), isn't $s_{t:t+c}$ inaccessible to the agent?
* Which variable is the Gaussian exploration noise added to, the decoded action sequence $u_t$, the latent variable $z_t$, or $v_t$?

**Limitations:**

The authors have stated in the Conclusion and Limitation section that representing long action sequences is a limitation and future direction of this work. However, there are no results suggesting that MARS suffers from long action sequences. It would be helpful to discuss more about this limitation.

---

> ### Author Rebuttal · Authors · 2024-08-06
>
> We are deeply thankful to you for recognizing the presentation and originality of our work; this positive feedback is greatly encouraging. Furthermore, your objective advice motivates us to further improve this work.
>
> If you think the following response addresses your concerns, we would appreciate it if you could kindly consider raising the score.
> ## Part 1
> ### Weaknesses
> **W1：The comparisons in the experiment are with some simple and intuitive baselines. While the authors mention that no specific solution exists for this problem, it would still be helpful to discuss related work in other research fields. It would be beneficial to include more tasks to validate the effectiveness of MARS.**
>
> Thanks for your valuable suggestions. We add multiple methods in similar fields as baselines and verify the effectiveness of all the methods in more complex scenarios than the original version.
>
> **Baselines:** We select the latest multi-step decision-making fully supervised method ACT [1] from the robotics learning area, which requires us to build an expert dataset for it via ppo in advance; and two recent SOTA methods in the Delayed MDP domain, i.e. DCAC [2], State Augmentation (SA) [3] and delayed Dreamer[4] (Relax the Intermitted setting restrictions, allow such methods to additionally use dense rewards and delay priors at each step, and set the delay coefficient to the interaction interval).
>
> **Benchmarks:** For simulation environments, we further select four more challenging DeepMind Control (DMC) tasks focused on bionic robot locomotion: Dog Run, Dog Trot, Dog Stand and Humanoid Walk. DMC tasks demand coordinated movement from multi-joint bionic robots. Besides, two robotic arm control tasks in MetaWorld: Sweep Into and Coffee Push are used.
>
> **Metrics:** We evaluate methods in terms of performance and Smoothness (whether the motion is coherent and stable, invalid jitter and stagnation will reduce the score, please refer to Sec.5.2 for detail).
>
> Table 1a. Performance score of random Intermitted MDP setting (smoothness (%)) (Average of 4 runs):
> | Method      |Dog Run|Dog Trot| Dog Stand| Humanoid Walk|Sweep Into|Coffee Push|
> | :-: | :-: | :-: | :-: | :-: | :-: | :-: |
> | **Ours**|$124.61\pm 44.92 (75)$|$574.12\pm 28.76 (88)$|$614.03\pm 17.42 (73)$|$105.47\pm 35.83 (82)$|$0.51\pm 0.13 (68)$|$0.38\pm 0.06(63)$|
> | ACT|$108.37\pm 36.51 (72)$|$476.95\pm 32.37 (79)$|$607.94\pm 20.56 (75)$|$110.71\pm 16.46 (76)$|$0.47\pm 0.07 (57)$|$0.21\pm 0.03 (69)$|
> | DCAC|$96.87\pm 28.44 (53)$|$426.93\pm 50.48 (72)$|$562.64\pm 22.73 (64)$|$105.32\pm 29.16 (49)$|$0.44\pm 0.27 (41)$|$0.19\pm 0.13 (48)$|
> | SA |$92.74\pm 51.06 (37)$|$385.67\pm 52.49 (39)$|$503.94\pm 14.86 (27)$|$75.66\pm 31.42 (45)$|$0.52\pm 0.21 (37)$|$0.34\pm 0.09 (39)$|
> | delayed Dreamer |$95.31\pm 26.74 (46)$|$428.39\pm 46.23 (46)$|$526.07\pm 21.84 (25)$|$89.25\pm 27.41 (41)$|$0.56\pm 0.17 (32)$|$0.39\pm 0.04 (37)$|
>
> Table 1b. Performance score of fixed Intermitted MDP setting (smoothness (%)) (Average of 4 runs):
> | Method      |Dog Run|Dog Trot| Dog Stand| Humanoid Walk|Sweep Into|Coffee Push|
> | :-: | :-: | :-: | :-: | :-: | :-: | :-: |
> | **Ours**|$162.52\pm 64.43 (82)$|$593.73\pm 23.13 (85)$|$622.68\pm 26.07 (79)$|$121.32\pm 52.16 (84)$|$0.64\pm 0.07 (73)$|$0.42\pm 0.11 (72)$|
> | ACT|$127.23\pm 29.33 (76)$|$493.56\pm 48.27 (87)$|$627.31\pm 14.83 (76)$|$103.21\pm 19.35 (78)$|$0.51\pm 0.09 (69)$|$0.34\pm 0.06 (64)$|
> | DCAC|$94.42\pm 19.24 (58)$|$451.92\pm 27.06 (79)$|$568.07\pm 38.26 (74)$|$113.73\pm 22.24 (54)$|$0.54\pm 0.05 (48)$|$0.28\pm 0.14 (52)$|
> | SA |$92.31\pm 26.36 (34)$|$405.78\pm 74.19 (43)$|$562.84\pm 34.69 (31)$|$83.73\pm 28.41 (55)$|$0.47\pm 0.13 (42)$|$0.29\pm 0.16 (44)$|
> | delayed Dreamer |$103.71\pm 73.49 (51)$|$485.93\pm 95.27 (56)$|$541.58\pm 39.46 (36)$|$94.61\pm 22.31 (52)$|$0.66\pm 0.13 (46)$|$0.43\pm 0.145 (48)$|
>
> Above results show that our method performs better than other methods in almost all intermitted MDP control tasks while ensuring smooth and coherent motion of the agent. The effect of the supervised learning method ACT outperforms the delayed MDP methods, and the delayed MDP methods perform well in the robotic arm scene but cannot maintain motion smoothness and time efficiency.
>
> **W2: Minors in writing**
>
> We apologize for any inconvenience caused by our rough writing. We refined our grammar, spelling, formulas, and notations.
> ### Questions
>
> **Q1: Which variable is the Gaussian exploration noise added to, the decoded action sequence $u_t$, the latent variable $z_t$, or $\upsilon_t$?**
>
> We add Gaussian perturbations to $z_t$ in this paper. Additionally, we analyzed each of the above methods on a random Intermitted Mujoco task and added it to the new submission.
>
> Table 2. Performance (Average of 4 runs):
> | Method | Ant-v2|
> | :-: | :-: |
> | $u_t$ perturbation|$5886\pm 242.68$|
> | $z_t$ perturbation |$5908\pm 194.61$|
> | $\upsilon_t$ perturbation |$5397\pm 206.84$|
>
> The above results indicate that adding noise to $u_t$ and $z_t$ yields comparable effects. Furthermore, experiments suggest that adding noise to $\upsilon_t$ has no good impact.
>
> [1]: Zhao, Tony Z., et al. "Learning fine-grained bimanual manipulation with low-cost hardware." RSS 2023.
>
> [2]: Ramstedt, Simon, et al. "Reinforcement learning with random delays." ICLR 2020.
>
> [3]: Nath, Somjit, et al. "Revisiting state augmentation methods for reinforcement learning with stochastic delays." CIKM 2021.
>
> [4]: Karamzade A, Kim K, Kalsi M, et al. "Reinforcement learning from delayed observations via world models."  PMLR 2024.
>
>
> **The second part of the response is in the following comment. We are grateful for your thorough and conscientious reviewing.**

---

> ### Author Response · Authors · 2024-08-06
> **Rebuttal by Authors**
>
> ## Part 2
> **Q2: Why does the input of the encoder include $s_{t:t+c}$? According to the description of the intermittent control task (Line 31: agents are unable to acquire the state sent by the executor), isn't $s_{t:t+c}$ inaccessible to the agent?**
>
> This question is quite detailed, and we provide further explanation in the updated version. During the policy training stage, only the decoder of the VAE is used to collaborate with the agent's policy training, and $s_{t:t+c}$ is not required in this phase. The encoder is only used in the pre-training stage.
>
> During the pre-training phase, the construction of the action space is self-supervised, meaning it is independent of policy learning （VAE only needs sufficient information about the environment). Thus, $s_{t:t+c}$ utilized by the encoder does not represent expert data gathered by the RL policy; instead, it is a randomly sampled augmented dataset (abundant but of low quality, please refer to Section 4.1 for detail).
>
> We set up a set of experiments in a random Intermitted MDP Mujoco scenario to compare the effectiveness of expert data (collected by ppo) and randomly generated data (random policy) for VAE training. Experimental results show that for a fixed action latent space, randomly sampled state transitions contain richer states (sufficient state transition in the environment can be sampled) and the constructed space is more effective.
>
> Table 3. Performance (Average of 4 runs):
> | Method      | Ant-v2|
> | :-----------: | :-----------: |
> | MARS with expert data|$5473\pm 219.33$|
> | **MARS with random data** |$5908\pm 194.61$|
> ### Limitations
> **L1: The authors have stated in the Conclusion and Limitation section that representing long action sequences is a limitation and future direction of this work. However, there are no results suggesting that MARS suffers from long action sequences. It would be helpful to discuss more about this limitation.**
>
> In the appendix of the new version, we tested MARS with varying interval step $c$ in the Walker2d environment. The other environmental parameters and network hyperparameters remained consistent with the main experiment.
>
> Table 4. Performance (Average of 4 runs):
> | Method      | c=4|c=8|c=12|c=16|c=20|c=24|
> | :-----------: | :-----------: |:-----------: |:-----------: |:-----------: |:-----------: |:-----------: |
> | MARS|$5309\pm 143.85$|$5283\pm 171.46$|$5309\pm 143.85$|$5194\pm 201.52$|$4758\pm 106.73$|$4514\pm 377.25$|
>
> The Above results reveal a diminishing performance of MARS as the decision step size increases beyond $c=20$. We attribute this trend to the limitations in representation capacity imposed by the MLP architecture in the VAE. In the future, we plan to investigate alternative effective networks like Transformers to enhance the construction capabilities of MARS within action spaces under very long interval step setting.

---

> ### Author Response · Authors · 2024-08-11
>
> Dear reviewer 5oZS, we sincerely apologize for the inconvenience caused by placing our important experimental results in the general PDF, which may have required additional time for you to locate them. To address this, We have strategically placed the experiment that should be added to the PDF onto the webpage (integrated with responses to each question) to enhance the readability.
>
> If you have any further questions or suggestions, please feel free to share them with us. Your feedback is invaluable and catalyzes enhancing our work.

---

> > ### Comment · Reviewer_5oZS · 2024-08-11
> >
> > I thank the authors for their detailed rebuttal. The additional experiments enrich and enhance the empirical results, and it would be beneficial to include them in the paper. I have updated my score accordingly.

---

### Official Review · Reviewer_TqM1 · 2024-07-15

**Soundness:** 2
**Presentation:** 3
**Contribution:** 2
**Rating:** 7
**Confidence:** 3

**Summary:**

This paper addresses the issue of intermittent control problems, common in real-world scenarios where interactions between decision-makers and executors are disrupted due to unstable communication channels. These disruptions lead to bidirectional blockages, preventing agents from acquiring state information and transmitting actions, thus reducing the efficiency of reinforcement learning (RL) policies. The paper models this problem as an Intermittent Control Markov Decision Process and proposes a solution called Multi-step Action Representation (MARS). MARS encodes a sequence of actions into a compact latent space, enabling RL methods to optimize smooth and efficient motion policies. The experiments demonstrate that MARS significantly enhances learning efficiency and performance in both simulation and real-world robotic tasks compared to existing baselines.

**Strengths:**

The paper is, in general, well written, except for some confusing notation use. The problem studied in the paper is important but under-explored in RL literature, which in my opinion, makes this paper significant.

**Weaknesses:**

Some presentations, especially on notation use, are unclear. See my questions. Section 4 is unnecessarily long and consists of a lot of redundant text.

**Questions:**

- Line 181: What's the upper limit of action change?
- Eq 3 defines how the action transition scale is computed, but in Figure 3, why does the policy need to output the action transition scale?
- You denote the reconstruction layer as $g_{\psi)1}$. What does "1" in the subscript mean? This is confusing since you use subscript to denote timestep as well. Similarly, what's the purpose "2" of $h_{\psi_2}$?
- Line 245: It's confusing to say "choosing optimal z" since you just sample z from a policy.
- It seems relevant to https://arxiv.org/pdf/2304.13705. How do you compare with MARS and this work?
- All the experiments in Figure 4 are conducted in the same time interval. How does the performance difference change over time interval?

**Limitations:**

Yes, it's discussed.

---

> ### Author Rebuttal · Authors · 2024-08-06
>
> We are deeply grateful for your recognition of our paper's motivation, performance, and potential academic impact. Your positive feedback is highly encouraging. We improved our work with your valuable questions.
>
> If you think the following response addresses your concerns, we would appreciate it if you could kindly consider raising the score.
> ## Part 1
> ### Questions
> **Q1: It seems relevant to "Learning Fine-Grained Bimanual Manipulation with Low-Cost Hardware" (https://arxiv.org/pdf/2304.13705). How do you compare with MARS and this work?**
>
> Thanks for sharing this related research, the paper you provided further motivates us to improve our work. And in the new submission, we discussed this question in detail.
>
> The difference between the two works (ours is referred to as MARS, and related work is referred to as ACT) :
>
> - **Focus:** ACT addresses cutting-edge challenges in robotics: Can learning enable low-cost and imprecise hardware to perform these fine manipulation tasks? However, our primary objective is to develop a plug-in module that enhances the RL algorithm's proficiency in Intermitted MDP tasks.
>
> - **Training style:** ACT is an end-to-end supervised training method, and MARS is an unsupervised training method.
>
> - **Form of application:** ACT, *a multi-step decision model*, primarily leverages the generative capability of C-VAE and depends on high-quality expert data to enhance the model's multi-step decision-making proficiency through imitation learning. MARS, *a multi-step action space construction model*, primarily utilizes the latent space construction capability of C-VAE to build the action space using low-quality random transition data. Subsequently, it aids in the training of RL-style training. Therefore, it is more suitable for scenarios where reinforcement learning excels.
>
> - **Technic:** ACT creatively introduces action chunking and temporal ensemble to address the compounding errors associated with imitation learning in a manner that aligns with pixel-to-action policies. MARS, on the other hand, assists in action space construction by introducing action transition scale and state residual guidance.
>
> Although there are significant differences between the two methods, ACT inspired us in two points:
>
> *We found ACT to be a dependable and valuable baseline, and we included it in the main experiments of the new submission.*
> - Based on the code provided in the paper, we migrated it to our setup. Initially, we utilized PPO to gather expert data for fully supervised training. We set the chunking number to the maximum interval for our task and configured the Temporal Ensemble to the recommended value in the paper,i.e. 4. Following the paper's suggestions, we trained using L1 loss.
> - Benchmarks: Consistent with the Mujoco tasks in the original version.
> - We find that ACT performs better than the original baselines we compare, but underperforms our method on most tasks.
>
> Table 1. Performance score of random Intermitted MDP setting (Average of 4 runs):
> | Method  |Ant-v2|Walker2d-v2| HalfCHeetah-v2| Hopper-v2|
> | :-----------: | :-----------: | :------------: | :-----------: | :-----------: |
> | **Ours**|$4354.61\pm 158.44$|$5436.42\pm 217.83$|$6175.63\pm 273.95$|$2613.58\pm 177.96$|
> | ACT|$3279.82\pm 127.83$|$4892.18\pm 383.07$|$5811.51\pm 108.33$|$2684 .31\pm 238.27$|
> | frameskip TD3|$492.53\pm 31.95$|$2584.18\pm 106.24$|$3281.45\pm 139.42$|$381.74\pm 53.67$|
> | Multi-step TD3|$408.25\pm 31.76$|$529.26\pm 68.21$|$112.73\pm 17.82$|$1173.93\pm 78.28$|
>
> *ACT inspired us to employ a transformer architecture (similar to a BERT-like training style) instead of an MLP to construct the C-VAE. This transition is expected to enhance the representation capabilities of MARS in future work.*
> - We conducted a set of experiments on the Ant-v2, and we observed that the transformer-based MARS shows promise in enhancing RL algorithms and shows a more significant increase in representation ability when the *interval time step* $c$ becomes longer.
>
> Table 2. Performance (Average of 4 runs):
> | Method      | c=4 | c=8 | c=12 |c=16 | c=20 |
> | :-----------: | :-----------: | :------------: | :-----------: |:-----------: |:-----------: |
> |Transformer based |$5281.771\pm 231.59$|$5417.26\pm 193.18$|$5513.47\pm 337.52$|$5604.31\pm 246.52$|$5337\pm 114.23$|
> | MLP based|$5309\pm 143.85$|$5283\pm 171.46$|$5309\pm 143.85$|$5194\pm 201.52$|$4758\pm 106.73$|
>
> **Q2: All the experiments in Figure 4 are conducted in the same time interval. How does the performance difference change over time interval?**
>
> Thanks for your valuable advice. In the appendix of the new version, we tested MARS with varying interval step $c$ in the Walker2d environment. The other environmental parameters and network hyperparameters remained consistent with the main experiment.
>
> Table 3. Performance (Average of 4 runs):
> | Method      | c=4|c=8|c=12|c=16|c=20|c=24|
> | :-: | :-: |:-: |:-: |:-: |:-----------: |:-----------: |
> | MARS|$5309\pm 143.85$|$5283\pm 171.46$|$5309\pm 143.85$|$5194\pm 201.52$|$4758\pm 106.73$|$4514\pm 377.25$|
>
> The Above results reveal a diminishing performance of MARS as the decision step size increases beyond $c=20$. We attribute this trend to the limitations in representation capacity imposed by the MLP architecture in the VAE. In the future, we plan to investigate alternative effective networks like Transformers to enhance the construction capabilities of MARS within action spaces under very long interval step setting.
>
> **Q3: Line 181: What's the upper limit of action change?**
>
> We covered this in detail in the new version. The upper action limit $B$ varies according to each task. The semantics is the maximum scale that an action can change. For example, The range of actions in mujoco is $[-1,1]$, then $B=1- (-1)$.
>
> **The second part of the response is in the following comment. We are grateful for your thorough and conscientious reviewing.**

---

> ### Author Response · Authors · 2024-08-06
> **Rebuttal by Authors**
>
> ## Part 2
> **Q4:Eq 3 defines how the action transition scale is computed, but in Figure 3, why does the policy need to output the action transition scale?.**
>
> This is an in-depth problem, and we emphasize it in the new version. In the second stage, the decoder is employed to decode the latent action selected by the policy, with the action transition scale (ATS) functioning as a condition term that dynamically adjusts to guide the decoder's generation process. This condition term helps constrain the latent variable within a smaller subspace to rectify any erroneous decisions made by the policy. Hence, ATS can be regarded as another decision space akin to the action latent space $z$. However, the ATS is not constructed through deep learning but according to our formulated approach in Sec. 4.1. To enable adaptive selection of ATS, which would be inefficient and costly to manually provide for each task or state, we leverage the adaptive decision-making capability of RL. This allows the policy to decide on actions within the latent space $z$ and allocate a separate decision head to select a number from ATS.
>
> This output structure has been extensively validated in the area of hybrid action space control [1][2]. In the new version, we have included ablation experiments on Ant-v2 to demonstrate the viability of entrusting the ATS selection of RL policy.
>
> The baseline method involves selecting appropriate ATS through pre-defined manual scripts without altering other modules. Results indicate that the RL policy can identify the suitable ATS and achieve commendable performance after a certain exploration step.
>
> Table 4. Performance (Average of 3 runs):
> | Method      |  training step = 50k|training step = 1m|training step = 2m|
> | :-----------: | :-----------: | :-----------: | :-----------: |
> | Ours |$1962\pm 421.42$|$3243\pm 109.25$|$4183\pm 171.46$|
> | Baseline |$2213\pm 311.97$|$3126\pm 128.36$|$4207\pm 136.61$|
>
> **Q5: Line 245: It's confusing to say "choosing optimal z" since you just sample z from a policy.**
>
> Thanks for pointing out our imprecise presentation. In the new version, "optimal" is removed.
>
> **Q6: You denote the reconstruction layer as $g_{\phi_1}$. What does "1" in the subscript mean? This is confusing since you use subscript to denote timestep as well. Similarly, what's the purpose "2" of $h_{\phi_2}$?**
>
> We apologize for any confusion caused by our presentation. We intend to use the number $i=\{1,2\}$ to represent the $i_{th}$ parallel output head (reconstruction layer) connected after the decoder. Redundant symbols are removed in the new version.
> ### Weakness
> **W1: Section 4 is unnecessarily long and consists of a lot of redundant text.**
>
>  In the latest version, we streamlined the content of Section 4 to enhance its conciseness and clarity.
>
> [1]: Li, Boyan, et al. "Hyar: Addressing discrete-continuous action reinforcement learning via hybrid action representation." ICLR 2022.
>
> [2]: Fan, Zhou, et al. "Hybrid actor-critic reinforcement learning in parameterized action space." IJCAI 2019.

---

> ### Comment · Reviewer_TqM1 · 2024-08-08
>
> Please use the rebuttal button to respond. The rebuttal has a 6000-character limit, but the official comments do not. Using comments to respond is inappropriate since this increases the burden of reviewing and is unfair to the other authors. I will skim through the response since it's too long.
>
> - Q1: From the comparison with ACT and MARS, I don't think you can draw a statistically significant conclusion that MARS is better than ACT since their mean scores are close and standard deviations are overlapped.
>   - Your response to "Focus" reads a bit weird. The first sentence talks about the goal of ACT paper. Yes, that's their goal, but I think in this rebuttal, you should discuss the "technical focus of the ACT model" instead of the goal of their paper.
>   - Training style: Got it.
>   - Form of application & Technique: I think you can apply ACT model to RL setting.
>
> - Q2: If I remember correctly, c is the max time interval of an MDP. What not doing this experiment on the other baselines?
>
> - Q3: Got it.
>
> - Q4: Please shorten it.
>
> - Q6: If you're talking about the output head in Figure 3, please consider the other subscript that represents the purpose of the output heads better.
>
> - In the latest version, we streamlined the content of Section 4 to enhance its conciseness and clarity: I need to see your detailed plan of revision; otherwise, I don't think the next version will be concise.
>
> I appreciate the author's additional experiments, but I will keep my rating since I still have lots of concern of letting this paper in (see my comments above).

---

> ### Author Response · Authors · 2024-08-10
> **Re-optimize and simplify our rebuttal**
>
> We apologize for using the comment button to rebuttal and the long reply due to an incorrect understanding of the rule (moving the pdf supplement to comment). In the future, we commit to rigorously adhering to the submission order.
>
> **Q1.1: A more convincing comparison is needed.**
>
> For this iteration, we raised the seed number to 8 and introduced six additional famous difficult scenarios (4 DeepMind Control scenarios: Dog Run, Dog Stand, Dog Trot, Humanoid Walk; 2 metaworld scenarios: Sweep Into, Coffee Push). interval step is $14$, mean advantage ratio: (Ours-ACT)/ACT
>
> The findings presented in Table 1 indicate that doubling the number of seeds leads to a notable reduction in score variance for each method, resulting in improved and more stable performance.
> Our method stands out prominently in all scenarios except Dog Trot and Coffee Push. Our analysis suggests that the random intervals in our scenario may be a contributing factor that restricts the performance of ACT.
>
> Table 1. Performance score of difficult tasks (Average of 8 runs):
> | Method|ACT|Ours| Mean advantage ratio (%)|
> | :-: | :-: | :-: | :-: |
> | Dog Run|$74.93\pm 31.26$|$124.61\pm 24.71$|$65.64$|
> | Dog Trot|$438.94\pm 36.27$|$574.12\pm 28.76$|$30.80$|
> | Dog Stand|$561.68\pm 11.34$|$626.11\pm 18.36$|$11.47$|
> | Humanoid Walk|$82.15\pm 19.34$|$108.82\pm 26.45$|$32.47$|
> |Coffee Push|$0.44\pm 0.07$|$0.53\pm 0.04$|$20.45$|
> |Sweep Into|$0.27\pm 0.03$|$0.39\pm 0.06$|$44.45$|
>
> **Q1.2: discuss the "technical focus of the ACT"**
>
> - Different from MARS, which is a lightweight auxiliary plug-in to improve the multi-step decision-making ability of the model-free RL methods, ACT directly constructs multi-step decision-making policy by imitation learning.
> - ACT uses Transformer-based Action Chunking and Temporal Ensemble **at the decision level** to ensure the stability of each step of the execution, i.e. obtains more accurate actions through the synthesis of $k$ decisions at the same time step.  In contrast, MARS focuses on developing lightweight plugins, utilizing MLP as the backbone, and incorporating action transition scale and state residual regularization terms to enhance the quality of action representation **at the VAE training level**.
>
> **Q1.3: You can apply ACT model to RL setting.**
>
> We successfully transitioned ACT into the RL setting by incorporating a momentum penalty into the standard reward and leveraging the Importance Experience Buffer.
>
> Adapting ACT to the RL setting showed a slight improvement over the original ACT in most scenarios. However, there remains a significant gap between ACT and MARS.
>
> Table 2. Performance (Average of 8 runs):
> | Method|Dog Run|Dog Trot| Dog Stand| Humanoid Walk|Sweep Into|Coffee Push|
> | :-: | :-: | :-: | :-: | :-: | :-: | :-: |
> | **MARS**|$124.61\pm 24.71$|$574.12\pm 28.76$|$626.11\pm 18.36$|$108.82\pm 26.45$|$0.53\pm 0.04$|$0.39\pm 0.06$|
> | RL setting ACT |$68.93\pm 31.62$|$462.32\pm 21.08$|$541.73\pm 14.86$|$89.12\pm 23.21$|$0.46\pm 0.05$|$0.31\pm 0.06$|
> | ACT |$74.93\pm 31.26$|$438.94\pm 36.27$|$561.68\pm 11.34$|$82.15\pm 19.34$|$0.44\pm 0.07$|$0.27\pm 0.03$|
>
> **Q2**
>
> We compare all methods under varied c settings on Walker2d-v2.
>
> Tab.3 shows that MARS exhibits minimal sensitivity to the length of the interval time step $c$, showcasing consistently outstanding performance. In the RL setting, ACT ranks second, but its performance decreases with longer interval steps. We aim to enhance ACT's online learning for extended intervals. original ACT struggles with stability in longer time steps. The other baselines are less effective, while frameskip shows better stability in longer interval scenarios.
>
> Table 3. Performance  (Average of 8 runs):
> | Method| Decision step $c=4$|$c=8$|$c=12$|$c=16$|
> | :-: | :-: |:-: |:- |:-: |
> | MARS |$5813\pm 126.52$|$5734\pm 203.05$|$5624\pm 115.31$|$5608\pm 142.67$|
> | RL setting ACT |$4792\pm 206.15$|$5176\pm 163.29$|$4136\pm 227.91$|$4092\pm 322.84$|
> | ACT |$4608.74\pm 148.92$|$4795.03\pm 163.28$|$3911.43\pm 215.53$|$3471.23\pm 186.31$|
> |frameskip-TD3|$3235.79\pm 107.42$|$3378.47\pm 203.28$|$3755.14\pm 291.07$|$3165.62\pm 108.76$|
> | multistep-TD3|$3766.31\pm 166.28$|$2904.01\pm 156.13$|$1007.43\pm 272.65$|$657.41\pm 75.85$|
>
> **Q4**
>
> RL policy requires the decoder of c-VAE to reconstruct the action sequence. During the reconstruction, the decoder must incorporate not only the latent action but also the suitable action transition scale (ATS) as a conditional term to ensure that the decoded action aligns with the current state.
> To enable adaptive selection of ATS, which would be inefficient and costly to manually provide for each task or state, we leverage the adaptive decision-making capability of RL (following the hybrid control RL output style). This allows the policy to decide on actions within the latent space $z$ and allocate a separate decision head to select a number from ATS.
>
> **Q6**
>
> We use $h_\theta$ and $g_\mu$ to represent the two output heads in the updated version.

---

> ### Author Response · Authors · 2024-08-10
> **Re-optimize the response to the weakness**
>
> **Show your detailed plan of revision**
>
> - In lines 164-180 of section 4.1 (Scale-conditioned Multi-step Action Encoding and Decoding), we condensed the repetitive introduction of the action transition scale. This streamlining is particularly evident in lines 173-178, where we illustrate with the example of robot motion,
> as we have already covered this concept in the Introduction.
> - Section 4.3 (DRL with Multi-step Action Representation) is redundant. We streamlined the redundant training process explanation (lines 267-277) since the preceding two subsections are described in detail and a pseudo-code is used to assist readers in comprehension.
>
> With the above reduction, the overall space of Section 4 is reduced by 3/4 pages, which just allows us to incorporate the newly added main experiment into the main text.
>
> *If you have any further questions or suggestions, please feel free to share them with us. Your feedback is invaluable and catalyzes enhancing our work.*

---

> > ### Comment · Reviewer_TqM1 · 2024-08-11
> >
> > Thanks for the clarification. I have no further questions and increased my rating.

---

### Author Rebuttal · Authors · 2024-08-06

## General Response
We sincerely appreciate all reviewers for their meticulous assessment and valuable insights to our paper. Special thanks to all three reviewers for their thorough and meticulous review of our submission.

Please permit us to present the additional primary experiment results here. Taking into account the recommendations from reviewer Bn1i and reviewer 5oZS, We have conducted a comparison of the performance of our method with several **additional baselines** chosen from the Delayed MDP and robotic learning domains across **more challenging scenarios.**

### Question: How does the method compare to related works in other areas, e.g. delay MDP methods?

We add a comparison of our method with mainstream delayed MDP methods on multiple complex tasks from the perspective of multiple metrics in the new version.

**Baselines:** We select several recent SOTA methods in the Delayed MDP domain, i.e. DCAC [1] and State Augmentation (SA) [2] (Relax the Intermitted setting restrictions, allow such methods to additionally use dense rewards and delay priors at each step, and set the delay coefficient to the interaction interval). For a more comprehensive analysis, a recent model-based approach,i.e. delayed Dreamer [3], is also chosen.

**Benchmarks:** We further select four more challenging DeepMind Control (DMC) tasks focused on bionic robot locomotion: Dog Run, Dog Trot, Dog Stand and Humanoid Walk. DMC tasks demand coordinated movement from multi-joint bionic robots. Besides, two robotic arm control tasks in MetaWorld: Sweep Into and Coffee Push are used. The environmental parameters and network hyperparameters remained consistent with the main experiment. For methods that require an expert dataset, we use the trained PPO to collect data in an ideal setting environment.

**Metrics:** We evaluate methods in terms of performance and smoothness (whether the motion is coherent and stable, invalid jitter and stagnation will reduce the score, please refer to Sec.5.2 for details).

Table 1. Performance in newly added difficult tasks (smoothness (%)) (Average of 4 runs):
| Method      |Dog Run|Dog Trot| Dog Stand| Humanoid Walk|Sweep Into|Coffee Push|
| :-----------: | :-----------: | :------------: | :-----------: | :-----------: | :-----------: | :-----------: |
| **Ours**|$124.61\pm 44.92 (75)$|$574.12\pm 28.76 (88)$|$614.03\pm 17.42 (73)$|$105.47\pm 35.83 (82)$|$0.51\pm 0.13 (68)$|$0.38\pm 0.06(63)$|
| DCAC|$96.87\pm 28.44 (53)$|$426.93\pm 50.48 (72)$|$562.64\pm 22.73 (64)$|$105.32\pm 29.16 (49)$|$0.44\pm 0.27 (41)$|$0.19\pm 0.13 (48)$|
| SA |$92.74\pm 51.06 (37)$|$385.67\pm 52.49 (39)$|$503.94\pm 14.86 (27)$|$75.66\pm 31.42 (45)$|$0.52\pm 0.21 (37)$|$0.34\pm 0.09 (39)$|
| delayed Dreamer |$95.31\pm 26.74 (46)$|$428.39\pm 46.23 (46)$|$526.07\pm 21.84 (25)$|$89.25\pm 27.41 (41)$|$0.56\pm 0.17 (32)$|$0.39\pm 0.04 (37)$|

Table 2. Performance in old tasks (smoothness (%)) (Average of 4 runs):
| Method  |Ant-v2|Walker2d-v2| HalfCHeetah-v2| Hopper-v2|
| :-----------: | :-----------: | :------------: | :-----------: | :-----------: |
| **Ours**|$4354.61\pm 158.44$ (78)|$5436.42\pm 217.83$ (82)|$6175.63\pm 273.95$ (76)|$2613.58\pm 177.96$ (83)|
| DCAC |$3279.82\pm 127.83$ (42)|$4892.18\pm 383.07$ (54)|$5811.51\pm 108.33$ (58)|$2684 .31\pm 238.27$ (63)|
| SA |$492.53\pm 31.95$ (51)|$2584.18\pm 106.24$ (59)|$3281.45\pm 139.42$ (66)|$381.74\pm 53.67$ (71)|
| delayed Dreamer|$408.25\pm 31.76$ (35)|$529.26\pm 68.21$ (63)|$112.73\pm 17.82$ (68)|$1173.93\pm 78.28$ (78)|

Table 1,2 show that our method performs better than Delayed MDP methods in almost all intermitted MDP control tasks while ensuring smooth and coherent motion of the agent.

*P.S. If there are any other Delayed MDP methods that you believe should be compared but fall outside the scope of our current investigation, please inform us, and we will gladly incorporate them into our experimental analysis.*


[1]: Ramstedt, Simon, et al. "Reinforcement learning with random delays." ICLR 2020.

[2]: Nath, Somjit, et al. "Revisiting state augmentation methods for reinforcement learning with stochastic delays." CIKM 2021.

[3]: Karamzade A, Kim K, Kalsi M, et al. "Reinforcement learning from delayed observations via world models." PMLR 2024.

---

### Decision · Program_Chairs · 2024-09-25

**Decision:**

Accept (poster)

**Comment:**

The paper addresses an important practical problem (intermittent loss of communication in  control systems) in a principled and interesting way. All reviewers are favourable and the authors did a great job during the rebuttal process of allaying concerns.